# Hit and Lead Discovery with Explorative RL and Fragment-based Molecule Generation

**Soojung Yang** *
AITRICS
soojungy@mit.edu

**Doyeong Hwang** †
AITRICS
desertbeagle11@gmail.com

**Seul Lee**
KAIST
ellenlee7890@gmail.com

**Seongok Ryu** ‡
AITRICS
seongokryu@galux.co.kr

**Sung Ju Hwang**
AITRICS, KAIST
sjhwang82@kaist.ac.kr

## Abstract

Recently, utilizing reinforcement learning (RL) to generate molecules with desired properties has been highlighted as a promising strategy for drug design. A molecular docking program – a physical simulation that estimates protein-small molecule binding affinity – can be an ideal reward scoring function for RL, as it is a straightforward proxy of the therapeutic potential. Still, two imminent challenges exist for this task. First, the models often fail to generate chemically realistic and pharmacochemically acceptable molecules. Second, the docking score optimization is a difficult exploration problem that involves many local optima and less smooth surfaces with respect to molecular structure. To tackle these challenges, we propose a novel RL framework that generates pharmacochemically acceptable molecules with large docking scores. Our method – Fragment-based generative RL with Explorative Experience replay for Drug design (FREED) – constrains the generated molecules to a realistic and qualified chemical space and effectively explores the space to find drugs by coupling our fragment-based generation method and a novel error-prioritized experience replay (PER). We also show that our model performs well on both *de novo* and scaffold-based schemes. Our model produces molecules of higher quality compared to existing methods while achieving state-of-the-art performance on two of three targets in terms of the docking scores of the generated molecules. We further show with ablation studies that our method, predictive error-PER (FREED(PE)), significantly improves the model performance.

## 1 Introduction

Searching for "hits", the molecules with desired therapeutic potentials, is a critical task in drug discovery. Instead of screening a library of countless potential candidates in a brute-force manner, designing drugs with sample-efficient generative models has been highlighted as a promising strategy. While many generative models for drug design are trained on the distribution of known active compounds [1–3], such models tend to produce molecules that are similar to that of the training dataset [4], which discourages finding novel molecules.

In this light, reinforcement learning (RL) has been increasingly used for goal-directed molecular design, thanks to its exploration ability. Previous models have been assessed with relatively simple

---

*Currently at MIT.
†Currently at LG AI Research.
‡Currently at Galux inc.

35th Conference on Neural Information Processing Systems (NeurIPS 2021).

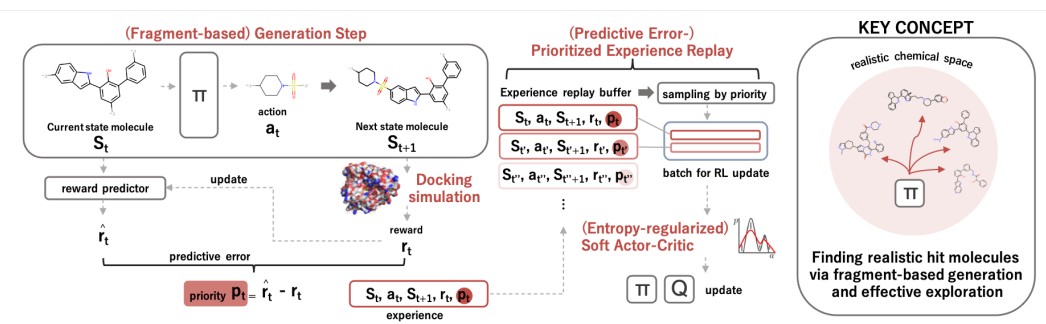

Figure 1: **Overview of our generative drug discovery method**. To find realistic 'hit' molecules that have high docking scores, we combine fragment-based generation method with SAC and PER. This figure illustrates our version of PER where the priority of the experience is defined by predictive error.

objectives, such as cLogP and QED, or estimated bioactivity scores predicted by auxiliary models [5–8]. However, high scores in those simple properties of molecules guarantee neither drug-likeness nor therapeutic potential, emphasizing the necessity of more relevant design objectives in generative tasks [9, 10]. The molecular docking method is a physical simulation that estimates the protein-small molecule binding affinity, a key measure of drug efficacy. As docking simulations are a more straightforward proxy of therapeutic potential, coupling RL with docking simulations would be a promising strategy. While the simplistic scores (e.g., cLogP, QED) are computed by a sum of local fragments' scores and are not a function of global molecular structure, making the optimization tasks relatively simple, docking score optimization is a more demanding exploration problem for RL agents. A change in docking score is nonlinear with the molecule's local structural changes, and a significant variance exists among the structures of high-scoring molecules, meaning that there exists many local optima [9].

In addition, many previous RL models often suffer from generating unreal and inappropriate structures. Docking score maximization alone is not sufficient to qualify molecules as drug candidates, as the molecules should satisfy strong structural constraints such as chemical realisticness and pharmacochemical suitability. In other words, small molecule drug candidates should have enough steric stability to arrive at the target organ in the intended form (chemical realisticness), and they should not include any seriously reactive or toxic substructure (pharmacochemical suitability). The low quality of generated molecules can arise from a single improper addition of atoms and bonds, which would deteriorate the entire sanity of the structure. Since such 'mistakes' are easy to occur, implicitly guiding the model (e.g., jointly training the model with QED optimization) cannot completely prevent the mistakes. Thus, explicitly restricting the generation space within realistic and qualified molecules by generating molecules as a combination of appropriate fragments can be a promising solution [1, 11, 12].

However, such a strong constraint in generation space would make the optimization problem of docking score even harder and render a higher probability of over-fitting to few solutions, urging a need for better exploration for RL agents. In this respect, we introduce a new framework, **Fragment-based generative RL with an Explorative Experience replay for Drug design (FREED)**, which encourages effective exploration while only allowing the generation of qualified molecules.

Our model generates molecules by attaching a chemically realistic and pharmacochemically acceptable fragment unit on a given state of molecules at each step[4]. We enforce the model to form a new bond only on the attachment sites that are considered as appropriate at the fragment library preparation step. These strategies enable us to utilize medicinal chemistry prior knowledge and successfully constrain the molecule generation within the chemical space eligible for drug design. We also explore several explorative algorithms based on curiosity-driven learning and prioritized

---

[4]In this work, we widely define the term "chemically realistic molecule" as a stable molecule and narrowly define it as a molecule that is an assembly of fragments that appear in the ZINC data. Also, we widely define the term "inappropriate molecule/fragment" or "pharmacochemically inacceptable molecule/fragment" as a molecule that has nonspecific toxicity or reactivity, and narrowly define it as a molecule that cannot pass through the three medicinal chemistry filters, which are Glaxo, PAINS, SureChEMBL filters.

experience replay (PER) [13]. We devise an effective novel PER method that defines priority as the novelty of the experiences estimated by the predictive error or uncertainty of the auxiliary reward predictor's outcome. With this method, we aim to avoid the lack of robustness of previous methods and encourage the exploration of diverse solutions. We provide an overall illustration of our framework in Figure 1.

Our main contributions can be summarized as follows:

- We propose a novel RL framework that can be readily utilized to design qualified molecules of high therapeutic potential.
- Our fragment-based generation method including connectivity-preserving fragmentation and augmentation allows our model to leverage Chemical prior knowledge.
- We propose novel explorative algorithms based on PER and show that they significantly improve the model performance.

## 2  Related Works

**SMILES-based and atom-based generation methods.**  SMILES-based methods [14] are infeasible for scaffold-based tasks[5] since molecular structures can substantially change through the sequential extension of the SMILES strings. Also, as explained in the Introduction, atom-based generation methods such as You et al.'s GCPN [6] inherently suffer from unrealistic generated molecules. Thus, we focus our discussion on motif-based generation methods.

**Motif-based molecular generation methods.**  A number of previous works [1, 11, 12] have investigated similar motif-based molecule generation based on the variational autoencoders (VAE). JT-VAE and HierVAE [1, 11] decompose and reconstruct molecules into a tree structure of motifs. These models might not be compatible with the scaffold-based generation, since a latent vector from their encoders depends on a motif-wise decomposition order which is irrelevant information for docking score that may bias the subsequent generation [12]. Maziarz et al. [12] also propose a VAE-based model which encodes a given scaffold with graph neural networks (GNNs) and decodes the motif-adding actions to produce an extended molecule. While Maziarz et al.'s motif-adding actions resemble our generation steps, we additionally introduce connectivity-preserving fragmentation and augmentation procedure which helps our model generate molecules of better quality.

Coupling an RL policy network with fragment-based generation holds general advantages compared to VAE-based methods. For example, RL models do not need to be trained on reconstruction tasks which might restrict the diversity of generated molecules. Our work is one of the earliest applications of RL with a fragment-based molecular generation method. While Ståhl et al.'s DeepFMPO [15] is also an application of RL with a fragment-based molecular generation method, DeepFMPO is designed to introduce only slight modifications to the given template molecules, which would make it inappropriate for the *de novo* drug design. Moreover, while DeepFMPO's generation procedure cannot change the connectivity of the fragments of the given template, our method is free to explore various connectivity.

**Docking as a reward function of RL.**  Studies on docking score optimization task has started very recently. Jeon et al. [16] developed MORLD, a atom-based generative model guided by MolDQN algorithm [17]. Olivecrona et al. [10] and Thomas et al. [18] utilized REINVENT[14], a simplified molecular-input line-entry system (SMILES)-based model generative model guided by improved REINFORCE algorithm [19] to generate hit molecules.

**RL algorithms for hard-exploration problems.**  Our view is that, on a high level, there are two main approaches to achieve efficient exploration. The first one is to introduce the "curiosity" or exploration bonus as intrinsic reward [20–22] for loss optimization. Bellemare et al. [23] first proposed a pseudo-count-based exploration bonus as an intrinsic reward. Pathak et al. [24] defined curiosity as a distance between the true next state feature vector and the predicted estimate of the next state feature vector. Thiede et al. [25] brought curiosity-driven learning into the molecular generation task. In Thiede et al., curiosity is defined as a distance between the true reward of the current state and the predicted estimate of the current state reward. However, these "curiosity-driven" intrinsic reward-based models sometimes fail to solve complex problems [26]. The failures are explained as a result of the RL agent's detachment or derailment from the optimal solutions [27].

---

[5]A drug design scheme where the drug candidates are designed from the given scaffold.

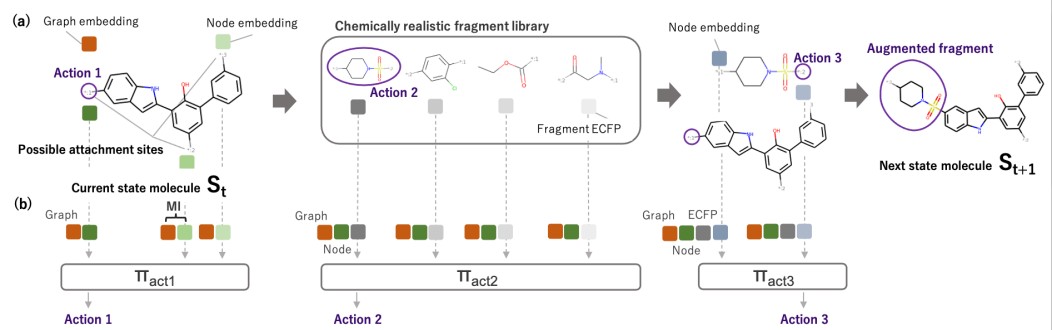

Figure 2: **Overview of our generation method (a) and policy network (b)**. Small colored squares represent a graph embedding of a molecule, node embedding of each attachment sitem and ECFP representation of each fragment. The graph or node embeddings and ECFP representations from the chosen actions are autoregressively passed onto the next action's policy network.

The other approach of solving hard exploration problems is a sample-efficient use of experiences [28, 13]. Prioritized experience replay (PER) method introduced in Schaul et al. [13] samples experiences that can give more information to the RL agent and thus have more 'surprisal' - defined by the temporal-difference (TD) error - in higher probability. In a similar sense, self-imitation learning (SIL) introduced in Oh et al. [29] samples only the 'good' experiences where the actual reward is larger than the agent's value estimate (i.e., estimate from the value function or Q function). However, prioritized sampling based on the agent's value estimate is susceptible to outliers and spikes, which may lead to destabilizing the agent itself [30]. Moreover, our explorative algorithm aims to preserve sufficient diversity and find many possible solutions instead of finding the most efficient route to a single solution. Thus, we modify the formulation of priority using the estimates for sample novelty.

## 3  Methods

### 3.1  Generation method

**Generation steps.**  The key concept of our method is to generate high-quality molecules that bind well to a given target protein. In order to achieve this goal, we devise a fragment-based generation method in which molecules are designed as an assemble of chemically realistic and pharmacochemically acceptable fragments. Given a molecular state, our model chooses "where to attach a new fragment", "what fragment to attach", and "where on a new fragment to form a new chemical bond" in each step[6]. Note that the action "where to form a new bond" makes our model compatible with docking score optimization since the scores depend on the three-dimensional arrangement of molecular substructures.

**Preserving fragment connectivity information in molecule generation.**  Another important feature of our method is harnessing the predefined connectivity information of the fragments and initial molecules. This feature allows our model to leverage chemists' expert knowledge in several ways. The connectivity information arises in the fragmentation procedure, in which we follow CReM [31] when the algorithm decomposes any arbitrary molecules into fragments while preserving the fragments' attachment sites as shown in Figure 3(a).

In the GNN embedding phase, the attachment sites are considered as nodes like any other atoms in the molecule, while an atom type is exclusively assigned to the attachment sites. We also keep tracking the indices of the attachment sites as the states so that our policy can choose the next attachment site where a new fragment should be added (Action 1). Similarly, we keep the indices of the attachment sites of the fragments and use them throughout the training so that our policy can choose the attachment site from the fragment side (Action 3).

Start and end points of new bonds are restricted to the attachment sites of given fragments and molecules. This restriction contributes to the chemical realisticness since the stability of the molecule

---

[6]Our model is designed to finish the episodes after four steps (in *de novo* cases) or two steps (in scaffold-based cases). At the end of the episode, the model substitute all the remaining attachment sites with hydrogen atoms.

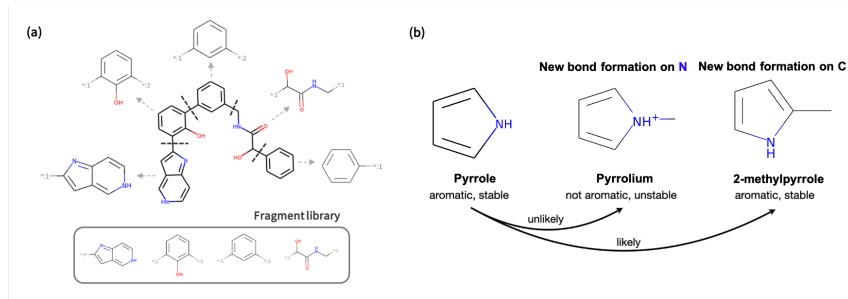

Figure 3: **Advantages of connectivity-preserving fragmentation. (a)** Substructure connectivity information is preserved as attachment sites in the fragmentation procedure. **(b)** Attaching a new bond to the nitrogen of a pyrrole ring would result in unstable molecule, breaking the aromaticity. Thus, pyrrole rings within an existing molecule are connected to other fragments by its carbon atoms, and preserving such an information would help us construct chemically realistic molecules.

depends on where the fragment is attached as illustrated in Figure 3**(b)**. Also, we can utilize our prior knowledge of protein-ligand interaction by rationally assigning the attachment sites of the initial molecule (scaffold), which have been widely harnessed by medicinal chemists in a usual scaffold-based scenario.

**Policy network.** Our generation method is coupled with the policy networks which guide the model to generate hit molecules. We provide an overall illustration of our policy network in Figure 2. Our state embedding network and policy network are designed as Markov models so that generation steps can take any arbitrary molecule as a current state and predict the next state, making the model more plausible for a scaffold-based generation.

In our framework, each state vector $s_t$ represents a molecular structure at the $t$-th step, and each step is a sequence of three actions that define an augmentation of a fragment. Following the method from Hwang et al. [32], the current state molecule is represented as an undirected graph $G$ which consists of a node feature matrix, an adjacency matrix, and an attachment site masking vector. A graph convolutional network (GCN) [33]-based network provides node embeddings, $H$, which are then sum-aggregated to produce a graph embedding, $h_g$. Note that the same GCN encodes the current state molecular graph (for Action 1) and the fragments (for Action 3).

Our policy network is an autoregressive process where Action 3 depends on Action 1 & 2 and Action 2 depends on Action 1.

$$p^{\text{act1}} = \pi_{\text{act1}}(Z^{\text{1st}}), Z^{\text{1st}} = \text{MI}(h_g, H_{\text{att}}) \tag{1}$$

$$p^{\text{act2}} = \pi_{\text{act2}}(Z^{\text{2nd}}), Z^{\text{2nd}} = \text{MI}(z^{\text{1st}}_{\text{act1}}, \text{ECFP}(h_{g_{\text{cand}}})) \tag{2}$$

$$p^{\text{act3}} = \pi_{\text{act3}}(Z^{\text{3rd}}), Z^{\text{3rd}} = \text{MI}(z^{\text{2nd}}_{\text{act2}}, U_{\text{cand}}) \tag{3}$$

where $H_{\text{att}}$ refers to the node embeddings of attachment sites. For the first step of our policy network, $\pi_{\text{act1}}$ takes the multiplicative interactions (MI) [34] of the node embedding of each attachment site and the rows of the graph embedding of the molecule, $z^{1st}_i \in Z^{1st}$, as inputs and predicts which attachment site should be chosen as Action 1. Since graph embeddings and node embeddings are defined in a heterogeneous space, we apply MI to fuse those two sources of information.

Given Action 1, $\pi_{\text{act2}}$ takes the MI of $z^{\text{1st}}_{\text{act1}}$, the row of $Z^{\text{1st}}$ under index $act1$, and the RDKit [35] ECFP (Extended Connectivity Fingerprint) representation of each candidate fragment, $h_{g_{\text{cand}}}$ as inputs, and predicts which fragment should be chosen as Action 2. By taking $Z^{\text{1st}}$ as one of the inputs, $\pi_{\text{act2}}$ reflects on the context of the current state and Action 1.

Finally, given Action 1 and 2, $\pi_{\text{act3}}$ takes the MI of $z^{\text{2nd}}_{\text{act2}}$ and the node embeddings of the chosen fragment's attachment sites, $U_{\text{cand}}$, as inputs and predicts which fragment attachment site should be chosen as Action 3. Each of the three policy networks $\pi_{\text{act1}}, \pi_{\text{act2}}$ and $\pi_{\text{act3}}$ consists of three fully-connected layers with ReLU activations followed by a softmax layer to predict the probability of each action. In order to make gradient flow possible while sampling discrete actions from the probabilities, we implement the Gumbel-softmax reparameterization trick [36].

## 3.2  Explorative RL for the discovery of novel molecules

**Soft actor-critic.**  We employ *soft actor-critic* (SAC), an off-policy actor-critic RL algorithm based on maximum entropy reinforcement learning [37, 38], which is known to explore the space of state-action pairs more effectively than predecessors. Assuming that our docking score optimization task requires more effective exploration than simplistic tasks, we chose SAC as our RL baseline algorithm.

$$\pi^* = \arg \max_{\pi} \sum_t \mathbb{E}_{(s_t, a_t) \sim \rho_\pi}[r(s_t, a_t) + \alpha \mathcal{H}(\pi(\cdot|s_t))] \tag{4}$$

$$\mathbb{E}_{s_t \sim \mathcal{D}}[\alpha \log \pi(a|s)] = \sum [\alpha \pi(a_t^{\text{act1}}, a_t^{\text{act2}}, a_t^{\text{act3}}|s_t)$$
$$\times (\log \pi(a_t^{\text{act1}}|s_t) + \log \pi(a_t^{\text{act2}}|s_t, a_t^{\text{act1}}) + \log \pi(a_t^{\text{act3}}|s_t, a_t^{\text{act1}}, a_t^{\text{act2}}))] \tag{5}$$

SAC aims to attain optimal policy that satisfies (4) where $\alpha$ is the temperature parameter balancing between exploration and exploitation of the agent, $\mathcal{H}(\pi(\cdot|s_t))$ is entropy of action probabilities given $s_t$, and $\rho_\pi$ is state-action transition distributions created from $\pi$. As we define act1, act2 and act3 autoregressively, the entropy regularization term defined in [39] $\mathbb{E}_{s_t \sim \mathcal{D}}[\alpha \log \pi(a|s)]$ is decomposed into (5).

**Explorative algorithms.**  To encourage exploration, we prioritize novel experiences during sampling batches for RL updates. We regard an experience as a novel experience if the agent has not visited the state before. Defining priority estimate function in the state space and not in the state-action space has been introduced for the molecular generative task in Thiede et al. [25]. For novel states, the reward estimator would yield a high predictive error or high variance (Bayesian uncertainty). In this regard, we train an auxiliary reward predictor consisting of a graph encoder and fully-connected layers that estimate a given state's reward (docking score). Then, we use the predictor's predictive error (L2 distance) or Bayesian uncertainty as a priority estimate. We name the former method **PER(PE)** and the latter method **PER(BU)**. The use of predictive error as a novelty estimate has been introduced for curiosity-driven learning [25], but our work is the first to apply this to PER.

For **PER(BU)**, we follow Kendall et al. [40] to obtain the Bayesian uncertainty of the prediction. We train the reward predictor network to estimate the reward's mean and variance (aleatoric uncertainty). Every layer in the network is MC dropout [41] layer so that the predictor can provide the epistemic uncertainty. We add aleatoric and epistemic uncertainty to obtain the total uncertainty of the estimate.

The reward predictor is optimized for every docking step, and we only optimize it based on final state transitions since docking scores are only computed for the final states. The reward predictor predicts the reward of any state, both intermediate and final. For **PER(PE)**, when we compute the priority of a transition including an intermediate state, we use Q value as a substitute for the true docking score. After updating the policy network and Q function with loss multiplied by importance sampling (IS) weight, we recalculate and update the priority values of the transitions in the replay buffer.

The following experiments section shows that PER with our priority estimate functions performs better than the previous methods.

## 4  Results and Analysis

### 4.1  Quantitative metrics

In this section, we introduce quantitative metric scores we used to assess our model. For every metric, we repeated every experiment five times with five different random seeds and reported the mean and the standard deviation of the scores. Also, we calculated the scores when 3,000 molecules were generated and used to update the model during training.

**Quality score.**  We report three widely used pharmacochemical filter scores – Glaxo [42], SureChEMBL [43], PAINS [44] – as quality scores. The quality scores are defined as a ratio of accepted, valid molecules to total generated molecules, as the filters reject the compounds that contain functional groups inappropriate for drugs (i.e., toxic, reactive groups). The higher the quality scores, the higher the probability that the molecule will be an acceptable drug. We also report the ratio of

valid molecules to total generated molecules (validity) and the ratio of unique molecules among valid generated molecules (uniqueness).

**Hit ratio.** We define hit ratio as a ratio of unique hit molecules to total generated molecules. We report the hit ratio to compare the model's ability to produce as many unique hit molecules in a given length of iterations, where we define *hits* as molecules whose docking scores are greater than the median of known active molecules' docking scores.

**Top 5% score.** We report the average score of the top 5%-scored generated molecules to compare the model's ability to produce molecules with better docking scores.

## 4.2 Quantitative performance benchmark

In this section, we compare the quality of the generated molecules and the model performance with three baseline models, MORLD [16], REINVENT [14], and HierVAE [11]. Our model, **FREED(PE)**, is our fragment-based generation method coupled with SAC and PER(PE). MORLD and REINVENT are the models utilized for docking score optimization tasks in previous works [16, 10, 18]. HierVAE is a strong non-RL fragment-based molecular generative model. Since HierVAE is a distributional learning method, we train HierVAE in two schemes - 'one-time' training on the known active molecules (**HierVAE**) and 'active learning (AL)' training where we train the model once on the known actives and twice on the top-scoring molecules from the generated molecules (**HierVAE(AL)**). In both schemes, we initialized the models with the model pre-trained on ChEMBL. We trained the models for three carefully chosen protein targets fa7, parp1, and 5ht1b. The choice of protein targets and the specifics of experimental settings are described in Appendix. For the experiments in this section, we use the small fragment library that includes 66 pharmacochemically acceptable fragments.

Table 1: **Quality scores of the models.** We trained our model and three baseline models with the target fa7 and computed quality scores of the first 3,000 molecules generated during training for each model. The two baseline models REINVENT and MORLD that are jointly trained to maximize filter scores are noted as 'REINVENT w/ filter' and 'MORLD w/ filter'. Standard deviation is given in brackets.

|  | Glaxo | SureChEMBL | PAINS | validity | uniqueness |
|---|---|---|---|---|---|
| MORLD | 0.561 (.009) | 0.131 (.013) | 0.805 (.013) | **1.000** (.000) | **1.000** (.000) |
| MORLD w/ filter | 0.578 (.010) | 0.145 (.018) | 0.816 (.008) | **1.000** (.000) | **1.000** (.001) |
| REINVENT | 0.773 (.023) | 0.667 (.030) | 0.769 (.022) | 0.813 (.024) | 0.988 (.008) |
| REINVENT w/ filter | 0.832 (.034) | 0.747 (.040) | 0.842 (.034) | 0.872 (.028) | 0.990 (.007) |
| HierVAE | 0.899 (.027) | 0.748 (.024) | 0.975 (.006) | 1.000 (.000) | 0.138 (.006) |
| HierVAE(AL) | 0.975 (.004) | 0.795 (.007) | 0.893 (.011) | 1.000 (.000) | 0.131 (.003) |
| Ours: FREED(PE) | **0.996** (.001) | **0.808** (.049) | **0.991** (.002) | **1.000** (.000) | 0.723 (.135) |

**Quality scores of generated molecules.** In real-world drug design, the generated molecules should be acceptable by pharmacochemical criteria. We excluded the fragments that are considered inappropriate by pharmacochemical filters to guarantee the quality of the generated molecules. Such an explicit constraint cannot be applied to atom-based (MORLD) or SMILES-based (REINVENT) methods.

As shown in Table 1, our model mostly generated acceptable molecules while the other models showed the poor rate of acceptable molecules, confirming our approach's advantage in drug design. We also investigated whether one could improve the quality of the molecules generated from the baselines by using a multi-objective reward function. MORLD and REINVENT optimized in multi-objective with both docking scores and the quality scores, denoted as 'MORLD w/ filter' and 'REINVENT w/ filter', show improved quality scores compared to single-objective MORLD and REINVENT. However, the multi-objective reward method was not as effective as our fragment-based approach, strengthening our claim that the explicit constraints are the effective strategy. Such a trend was consistent for all three targets. See Appendix for visual inspection of the quality score results.

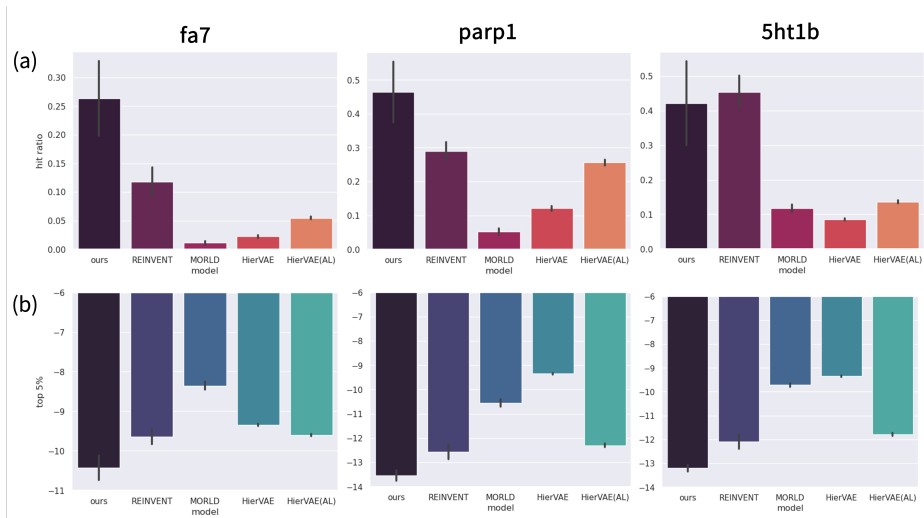

Figure 4: **Hit ratio and top 5% score of our model FREED(PE), REINVENT, MORLD, Hier-VAE, and HierVAE(AL).** Standard deviation is given as error bars. Higher hit ratios and greater negative value of the top 5% scores indicate better performance.

HierVAE showed high quality scores, as the HierVAE fragment library itself had very few problematic substructures (See Appendix A.5 for details). Such a result substantiates the advantage of the explicit fragment-based approach. However, HierVAE showed low uniqueness, possibly due to the small size of training data for fine-tuning (~1,200 known active molecules) and active learning (~1,500 generated high-scoring molecules).

We provide quality scores of our model (FREED(PE)) trained with the small library and the large library for all three protein targets in Table 2 and Table 3 of Appendix A.1, respectively. A significant increase in uniqueness was observed when we used the large library, which implies that the low uniqueness of our model is due to the small size of the fragment library. Thus, we believe constructing a fragment library that is large enough to guarantee high uniqueness while only including pharmacochemically acceptable fragments will be the best strategy in production.

**Docking score optimization result.**  Figure 4 shows the hit ratios and top 5% scores of the generative models. Our model outperforms the other generative models MORLD, REINVENT, and HierVAE in terms of both hit ratio and top 5% score, except for the hit ratio of the 5ht1b case. Such results show that our model's performance is superior or at least competitive to existing baselines, while our model exhibits many more practical advantages such as generating acceptable molecules, integrating chemist's expert knowledge by connectivity-preserving fragmentation and augmentation, and the feasibility in both *de novo* and scaffold-based generation.

### 4.3   Ablation studies: explorative algorithms

We also perform ablation studies of our algorithm to investigate the effect of our explorative algorithms. We used the larger library of 91 unfiltered fragments, as this section assesses the effect of the algorithms on the model's performance regardless of the pharmacochemical acceptability.

In Figure 5, we observe that all SAC models with explorative algorithms performed better than the vanilla SAC model, while the PPO model showed the worst performance. FREED with both PE and BU outperformed curiosity-driven models with PE and BU, showing the effectiveness of our methods in our task compared to curiosity-driven explorations. Moreover, our predictive error-PER method outperformed the TD error-PER method. We conjecture that such a result is due to 1) novelty-based experience prioritization encourages better exploration 2) leveraging an auxiliary priority predictor network makes PER training more robust than internal value estimate functions (Q function). We provide the significance analysis (one-tail paired t-test) of the result in Table 4 of Appendix A.1.

### 4.4   Case study on drug design

In this section, we show the practicality of our framework on *de novo* and scaffold-based drug design. We test FREED(PE) with our large fragment library which includes 91 fragments.

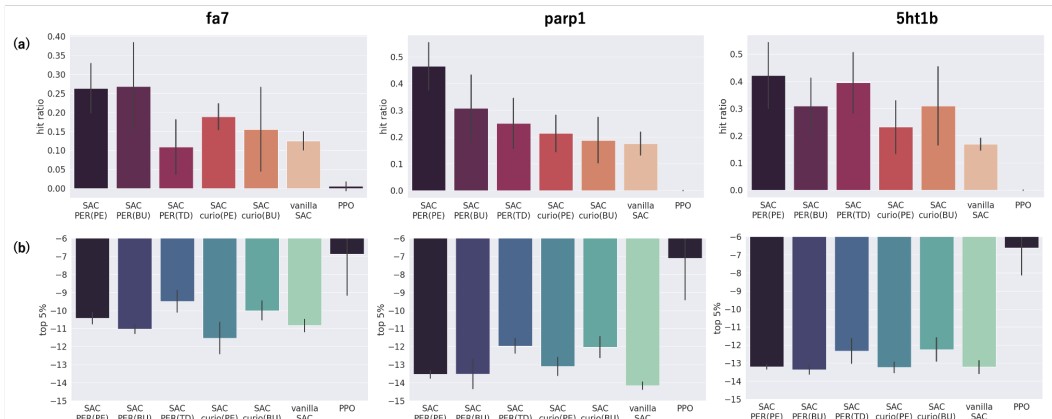

Figure 5: **Hit ratio and top 5% score of ablation studies.** Models can be categorized by whether they use {PER, curiosity-driven exploration(curio)}, and whether they use {predictive error from predictor(PE), Bayesian uncertainty(BU), and TD error from agent(TD)} as priority or intrinsic reward. Standard deviation is given as error bars.

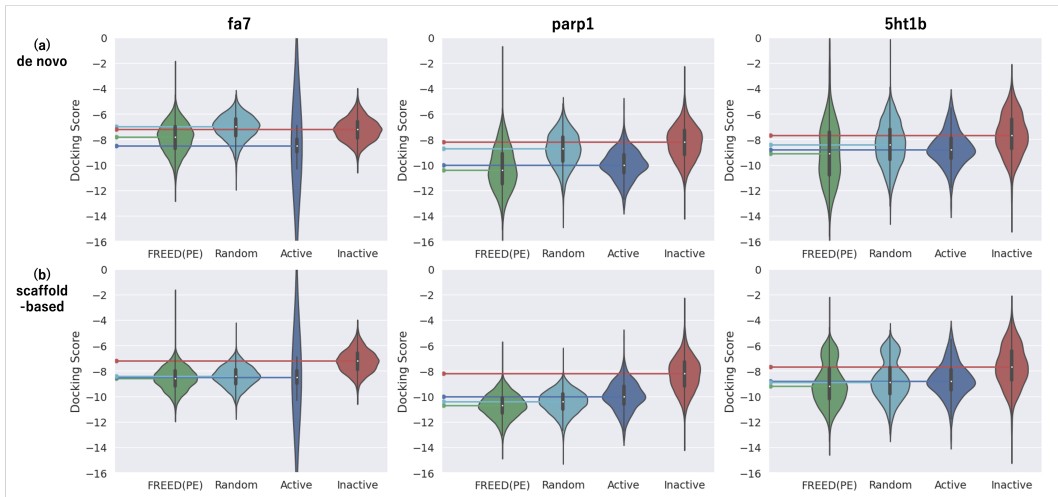

Figure 6: **Docking score distribution of the generated molecules.** Duplicate molecules were removed after gathering 3,000 molecules each from five random seed experiments. "Random" molecules are generated by our fragment-based generation method without training the policy network. "FREED(PE)" molecules are generated by the fragment-based generation method while training the policy network. We also plot known "Active" and "Inactive" molecules from DUD-E (fa7, parp1) or ChEMBL (5ht1b) datasets for comparison. Colored horizontal lines indicate the median of the corresponding distribution. **(a)** *de novo* scenario **(b)** scaffold-based scenario

***De novo* scenario.** Figure 6 **(a)** shows the distribution of the generated molecules before ("random") and after ("FREED(PE)") optimizing the policy network. Our model was able to effectively generate molecules that have higher docking scores compared to the known active molecules. Figure 7 **(i)** shows the structure of each target's optimized molecules.

**Scaffold-based scenario.** We validate our model on a scaffold-based scenario, where we attempt to improve docking scores by adding fragments to an initial scaffold molecule. Figure 6 *(b)* shows the distribution of the optimized molecules before ("random") and after ("FREED(PE)") training the policy network, with a scaffold of each target as an initial molecule.
Figure 6 (b) highlights our model's ability to optimize a given scaffold to have a higher binding affinity with the target. Surprisingly, in Figure 6 (b), even the molecules randomly optimized with our fragment-based generation algorithm show high docking scores when given the proper scaffold.

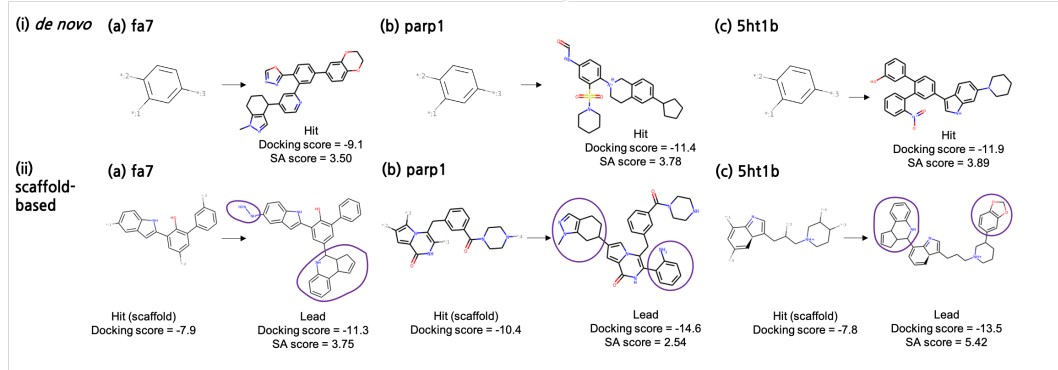

Figure 7: **Generated samples and their docking scores with our method, for *de novo* (i) and scaffold-based scenario (ii).** For each target, one of the high-scoring generated molecules is displayed with the initial molecule (benzene ring or scaffold). The purple line highlights the fragments augmented by the model in a scaffold-based generation. Numbers below the compounds are the docking scores and SA scores.

This result implies the importance of scaffold in hit discovery and highlights our generative method's ability to to span the chemical space around the scaffold effectively.

Figure 7 **(ii)** shows the structure of each target's scaffold and corresponding optimized molecules. We can see that the scaffold structures are well preserved in generated lead molecules. We provide an analysis of 3D docking poses of the scaffolds and generated lead molecules in Figure 2 and Figure 3 of Appendix A.1.

It is notable that our framework does not have to be specific for *de novo* or scaffold-based scenarios, except for the initial molecule and number of fragments to be added. Since our model is fully Markovian, whether the initial molecule is a benzene ring or a scaffold does not affect the model's training.

**Chemical realisticness of generated molecules.** In Figure 7, we report the SA (synthetic accessibility) score of the molecules, which is a widely used metric that estimates ease of synthesis by penalizing the presence of non-standard structural features. The SA score distribution of the catalogue molecules of commercial compound providers has its mode around 3.0 [45]. Accordingly, we can assume our generated molecules as reasonably synthesizable and thus chemically realistic.

## 5   Conclusion

In this work, we developed FREED, a novel RL framework for real-world drug design that couples a fragment-based molecular generation strategy with a highly explorative RL algorithm to generate qualified hit molecules. Our model generates pharmacochemically acceptable molecules with high docking scores, significantly outperforming previous docking RL approaches. Our code is released at https://github.com/AITRICS/FREED.

**Limitations and future work.** While our method does not explicitly account for the synthesizability of generated molecules, we believe forward synthesis-based methods [46] can be complementary to ours. It would be able to combine our method and forward synthesis-based method by substituting our attachment site bond formation actions with chemical reactions. In this way, we can explicitly take synthesizability into account while providing the appropriate model inductive bias for docking score optimization. We leave such an improvement as future work.

**Negative societal impacts.** If used maliciously, our framework can be utilized to generate harmful compounds such as biochemical weapons. Thus, conscious use of the AI model is required.

## Acknowledgments and Disclosure of Funding

We thank anonymous reviewers, Seung Hwan Hong, and Sihyun Yu for providing helpful feedback and suggestions. This work was supported by the National Research Foundation of Korea (NRF) grant funded by the project NRF2019M3E5D4065965.

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
