# A Appendix

## A.1 Additional experimental results

We further introduce our additional experiments in this section.

**Training baseline models jointly with pharmacochemical filter scores.** In our main article, we compared our model FREED with baseline models REINVENT and MORLD. For fairer comparison of quality scores, we also performed multi-objective optimization of REINVENT and MORLD on both quality score (pharmacochemical filter score) and docking score as follows.

$$\text{total reward} = \text{docking score reward} + \text{pharmacochemical filter reward} * 0.5 \qquad (1)$$

Pharmacochemical filter scores, calculated as 1 if accepted and 0 if rejected by each filter, were multiplied by 0.5 and then added to docking score reward (docking score reward $= -$docking score) to obtain the total reward.

Table 1 in the main text shows that such an implicit method is not enough to achieve nearly perfect filter scores as our model did. Also, as shown in Table 1 REINVENT showed deteriorated performance when jointly trained with filter scores, in terms of hit ratio and top 5% scores, implying that multi-objective optimization is more difficult than explicitly constrained optimization. Such a result was consistent for all three targets.

Table 1: Performance scores of the models. The two baseline models REINVENT and MORLD that are jointly trained to maximize filter scores are noted as REINVENT w/ filter and MORLD w/ filter. Standard deviation is given in brackets.

|  | hit ratio | top 5% score |
|---|---|---|
| MORLD | 1.1% (0.3%) | -8.353 (.105) |
| MORLD w/ filter | 1.8% (0.6%) | -8.483 (.148) |
| REINVENT | 11.8% (2.6%) | -9.649 (.186) |
| REINVENT w/ filter | 8.8% (2.5%) | -9.391 (.160) |
| Ours: FREED(PE) | **26.3%** (6.5%) | **-10.426** (.314) |

To show that the larger library allows our model to generate more unique molecules, we provide quality scores of our model (FREED(PE)) trained with the small library and the large, unfiltered library in Table 2 and Table 3.

Table 2: Quality scores of our model (FREED(PE)) trained with the small library (number of fragments = 66). We trained our model with all three targets and computed quality scores of the first 3,000 molecules generated during training. Standard deviation is given in brackets.

|  | Glaxo | SureChEMBL | PAINS | validity | uniqueness |
|---|---|---|---|---|---|
| fa7 | 0.996 (0.001) | 0.808 (0.049) | 0.991 (0.002) | 1.000 (0.000) | 0.723 (0.135) |
| parp1 | 0.995 (0.002) | 0.854 (0.050) | 0.991 (0.010) | 1.000 (0.000) | 0.557 (0.141) |
| 5ht1b | 0.995 (0.005) | 0.823 (0.106) | 0.990 (0.007) | 1.000 (0.000) | 0.592 (0.243) |

Table 3: Quality scores of our model (FREED(PE)) trained with the large library (number of fragments = 91). We trained our model with all three targets and computed quality scores of the first 3,000 molecules generated during training. Standard deviation is given in brackets.

|  | Glaxo | SureChEMBL | PAINS | validity | uniqueness |
|---|---|---|---|---|---|
| fa7 | 0.754 (0.072) | 0.558 (0.094) | 0.623 (0.201) | 1.000 (0.000) | 0.914 (0.162) |
| parp1 | 0.717 (0.167) | 0.577 (0.160) | 0.490 (0.169) | 1.000 (0.000) | 0.827 (0.160) |
| 5ht1b | 0.690 (0.204) | 0.551 (0.200) | 0.484 (0.138) | 1.000 (0.000) | 0.801 (0.191) |

Table 4: **Significance analysis of the results presented in Figure 5 in the main text.** One-tail paired t-tests were performed with the null hypothesis of $m_1 \leq m_2$, where $m_1$ is the score from the compared model and $m_2$ is the score from the vanilla SAC model.

|       | PER(PE) | PER(BU) | PER(PD) | curio(PE) | curio(BU) |
|-------|---------|---------|---------|-----------|-----------|
| fa7   | 0.00510 | 0.03410 | 0.65375 | 0.00910   | 0.30790   |
| parp1 | 0.00060 | 0.05000 | 0.09800 | 0.18900   | 0.39950   |
| 5ht1b | 0.00660 | 0.02480 | 0.00713 | 0.13550   | 0.06150   |

**Significance analysis of the ablation study results.** We performed one-tail paired t-tests on the results presented in Figure 5 in the main text. We compared the models' performances in terms of hit ratio with the vanilla SAC model. The result is shown in Table 4. From the $p < 0.05$ standard, we can see that FREED(PE) and FREED(BU) show significantly better results compared to the vanilla SAC model. Notably, the comparison with FREED(PE) results in a p-value of 0.0066, even below the 0.01 standard.

**Fragments from known active compounds.** Another advantage of our fragment-based generation method is that we can select our own fragment library. In this experiment, molecules were randomly generated until the 2,000th iteration step of our generation algorithm. Molecules were constructed from two different sets of fragments – "active" fragment library (see Figure 10), a set of fragments extracted from known DUD-E active compounds of fa7 target, and "large" fragment library, a set of fragments extracted from random ZINC molecules which were also used for experiments in Section 4.3 and Section 4.4. In Figure 1, active fragments-generated molecules show higher docking scores than random fragments-generated molecules. Also, while the average hit ratio was 7.24 % for random fragments-generated molecules, the average hit ratio of active fragments-generated molecules was 18.48 %, meaning that finding the potential hits becomes more than two times easier just by reconstructing molecules from the fragments of the known active compounds.

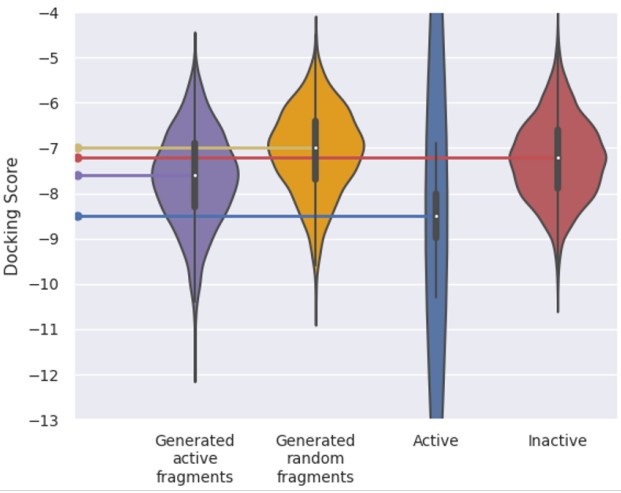

Figure 1: **Docking score distributions of the molecules generated with our method**. Molecules randomly generated from "active fragments" and "random fragments" are depicted as purple and yellow, respectively. Known active compounds and inactive decoys from DUD-E are depicted as blue and red, respectively.

**Docking pose analysis of generated molecules** In this analysis, we compare the 3D docking poses of the scaffolds and the generated leads in the Figure 7 of the main text. The 3D PyMOL [1] images describe fa7(left), parp1(middle), and 5ht1b(right) binding with their scaffolds and the generated molecules based on those scaffolds.

Firstly, for fa7, the binding poses of the scaffold and the generated molecules almost overlap. Such an overlap implies that the generated molecule will have a high binding affinity in high confidence.

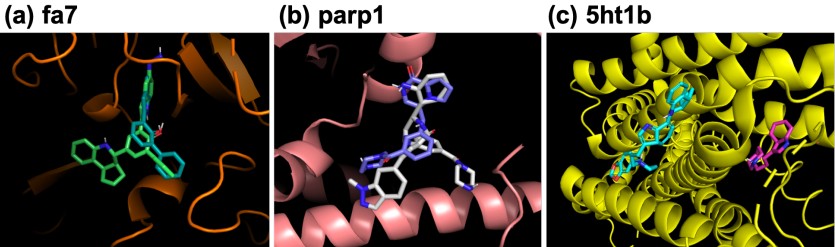

Figure 2: **3D PyMOL images of the binding poses of the scaffolds and the generated lead molecules.**

We profiled the details of the protein-ligand interactions of the scaffold and the generated molecule with a popular tool PLIP [2].

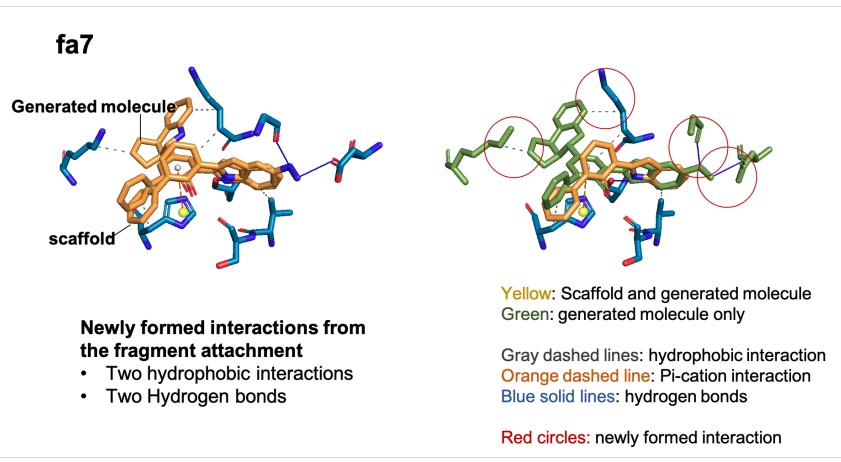

Figure 3: **PLIP image of the binding pose of the fa7 scaffold and the generated lead molecule.**

All existing interactions in the scaffold were preserved for the scaffold-based generated molecule, while several interactions were additionally formed. In detail, two hydrophobic interactions and two hydrogen bonds were formed by the augmented fragments.

For parp1, the scaffold and the generated molecule do exist in the same binding site, but the generated molecule has rotated about 180 degrees from the scaffold. Lastly, for 5ht1b, the scaffold and the generated molecule are docked in different binding sites. Since the generated molecule of 5ht1b is twice the size of the 5ht1b scaffold, we assume that the generated molecule could not fit in the original binding pocket.

Those three examples of generated molecules were randomly chosen among the high-scoring generations. Coincidentally, the three examples show us the possible orientation and binding site changes cases in scaffold-based generations. Since the changes in orientation and binding site affect the fidelity of the optimization, preventing such changes would be the future direction of our work.

**Scalability experiments.** In this experiment, we tested our model's performance on the larger action space. We constructed a fragment library of 350 fragments and a fragment library of 1k fragments and trained our model on both libraries. Table shows the performance and uniqueness score.

Although our model achieved lower performance scores with larger fragment libraries, the scores are better than or comparable to the baseline models. Moreover, as expected, the models trained with the larger libraries achieved much higher uniqueness with larger fragment libraries. Further investigation on the appropriate size of the fragment library considering the chemical space coverage and model performance would help the practitioners, and we leave this as our future work.

Table 5: Performance scores of the models using larger fragment libraries. The number of fragments in the libraries is denoted in parenthesis. Standard deviation is given in brackets.

|  |  | hit ratio | top 5% score |
|---|---|---|---|
| **fa7** | FREED(PE) (n=91) | 26.3% (6.5%) | -10.426 (.314) |
|  | FREED(PE) (n=350) | 9.3% (0.5%) | -9.179 (.053) |
|  | FREED(PE) (n=1k) | 10.1% (0.7%) | -9.259 (.053) |
|  | MORLD | 1.1% (0.3%) | -8.353 (.105) |
|  | REINVENT | 11.8% (2.6%) | -9.649 (.186) |
|  | HierVAE(AL) | 5.5% (0.2%) | -9.598 (.223) |
| **parp1** | FREED(PE) (n=91) | 46.4% (8.9%) | -13.537 (.219) |
|  | FREED(PE) (n=350) | 30.5% (1.2%) | -11.788 (.046) |
|  | FREED(PE) (n=1k) | 31.3% (1.4%) | -11.788 (.071) |
|  | MORLD | 5.2% (0.9%) | -10.551 (.143) |
|  | REINVENT | 29.0% (0.3%) | -12.565 (.298) |
|  | HierVAE(AL) | 25.6% (0.9%) | -12.306 (.414) |
| **5ht1b** | FREED(PE) (n=91) | 42.1% (12.1%) | -13.196 (.139) |
|  | FREED(PE) (n=350) | 57.9% (0.4%) | -11.746 (.048) |
|  | FREED(PE) (n=1k) | 60.3% (0.5%) | -11.751 (.023) |
|  | MORLD | 11.8% (1.1%) | -9.715 (.073) |
|  | REINVENT | 45.4% (0.5%) | -12.094 (.301) |
|  | HierVAE(AL) | 13.6% (0.4%) | -11.779 (.353) |

**Distributional analysis of generated molecules. 1) :** We investigate the coverage of chemical space that our large fragment vocabulary provides. We compare the distribution of 10.5k randomly generated molecules from our model and 2.5k ZINC molecules that we utilized to generate the large fragment library. We performed 2-dimensional uniform manifold approximation and projection (UMAP) on the Morgan fingerprints of the molecules (Figure ). Figure shows that our 10.5k randomly generated molecules almost completely cover the area 2.5k ZINC molecules occupy. Interestingly, our generated molecules' distribution expands radially from its center, where most ZINC molecules are located. Hence, it is implied that the chemical space coverage of generated molecules heavily depends on their 'parents' - the molecules that were fragmented to compose a fragment library. Thus, it will be optimal to take as many diverse ZINC molecules for fragmentation to improve the coverage.

**2) Chemical space coverage depending on fragment library generation method:** In the fragment library generation procedure, if two fragments with the same graph structure have different attachment sites, we removed the one that has fewer attachment sites. Alternatively, we can 'set sum' the all the attachment sites for a fragment. We test the alternative scenario and compare the chemical space coverage of the two methods.

**3) Chemical space coverage depending on the size of the fragment library:** As we perform the scalability experiments, we also investigate the chemical space coverage of larger fragment libraries (n=1k).

## A.2   Algorithm implementations

This section refers to the original works of the algorithms we have employed and elaborates on our implementation.

**Soft Actor-Critic.**   We employ SAC as implemented in OpenAI *spinningup*[1] [3]. Each action is predicted by corresponding policy network.

**Multiplicative Interaction.**   Jayakumar et al. [4] proposed multiplicative interaction (MI) as a powerful method to fusing information from multiple streams and showed that MI could be an alternative to basic concatenation calculation with better performances. We replace basic concatenation operation into MI in the case when fusing two vectors from a different scope. For example, MI is used in our

---

[1]MIT License, Copyright (c) 2018 OpenAI (http://openai.com)

policy network when fusing information from a node embedding vector and a graph embedding vector, in order to express the node vector's information with respect to the given graph vector.

**Gumbel-softmax.** In terms of implementation, sampling a single action does not require gradient flow in the original SAC. However, in our policy network, the actions are autoregressively defined. In detail, sampling Action 1 gives us the embeddings that will be used as an input for Action 2 policy, and likewise, sampling Action 2 gives the embeddings for Action 3 policy. Action 1, 2, and 3 together are considered a transition since the next state can only be reached after performing all three actions. As a continual gradient should flow through those autoregressive actions, we employed the Gumbel-softmax reparameterization trick [5] to replace argmax operations.

We modify the original Gumbel-softmax formula with an additional ratio multiplied to Gumbel distribution as $\nu$:

$$y_i = \frac{\exp((\log(\pi_i) + \nu \cdot g_i)/\tau)}{\sum_{j=1}^{k} \exp((\log(\pi_j) + \nu \cdot g_j)/\tau)}, g_i \sim \text{Gumbel}(0, 1). \tag{2}$$

$\nu$ is set to $10^{-3}$ and $\tau$ is set to $10^{-1}$.

**Prioritized experience replay.** Thrun et al. [6] and Oh et al. [7] argue that exploiting important and meaningful experiences would improve the performance in difficult exploration problems. We leverage prioritized experience replay(PER) method [8] to encourage exploration in our framework.

PER is a method that prioritizes experiences by replaying the important transitions more frequently. Probability of sampling transition $i$ is defined as

$$P(i) = \frac{p_i^{\alpha}}{\sum_k p_k^{\alpha}} \tag{3}$$

where $p_i > 0$ is the priority of $i$th transition, and the exponent $\alpha$ controls how much prioritization is used. In order to handle bias introduced from prioritized sampling, importance sampling(IS) weight $w_i$ is defined as

$$w_i = (\frac{1}{N} \cdot \frac{1}{P(i)})^{\beta} \tag{4}$$

and is multiplied to the loss defined in SAC.

**Curiosity-driven learning.** Curiosity-driven exploration methods direct the agent towards exploratory directions by introducing intrinsic reward that reflects on the 'surprisal' or novelty of a state. The intrinsic reward is then added to the external reward given by the environment to provide the total reward for the model update.

Intrinsic reward of the state is defined as absolute value of predictive error of the reward predictor $r_i^{\text{intr}} = |\hat{y}_i - r_i|$. The reward predictor is separately optimized to minimize MSE loss of predicted value and actual reward value:

$$\mathcal{L}_{\text{intr}}(\theta) = \sum_i (\hat{y}_i - r_i)^2 \tag{5}$$

**Bayesian uncertainty.** Kendall et al. [9] presented a Bayesian deep learning method estimating epistemic and aleatoric uncertainty. Monte-Carlo dropout [10] is used to produce the mean and variance of the predicted value of the given $i$-th input. Predictive uncertainty for a data point $y$ can be approximated by:

$$\text{Var}(y_i) \approx \frac{1}{T} \sum_{t=1}^{T} \hat{y}_{i,t}^2 - (\frac{1}{T} \sum_{t=1}^{T} \hat{y}_{i,t})^2 + \frac{1}{T} \sum_{t=1}^{T} \hat{\sigma}_{i,t}^2 \tag{6}$$

with $\{\hat{y}_{i,t}\}_{t=1}^{T}$ as a set of $T$ sampled outputs, where $\hat{y}_{i,t}, \hat{\sigma}_{i,t}^2 = \boldsymbol{f}^{\hat{W}_t}(x_i)$ for random dropout weights $\hat{W}_t \sim q(W)$. Estimated uncertainty in (6) is known to capture epistemic and aleatoric uncertainty. The loss function for the Bayesian neural network is defined as:

$$\mathcal{L}_{\text{BNN}}(\theta) = \frac{1}{D} \sum_i^{\frac{1}{2\hat{\sigma}_i^2}} \|y_i - \hat{y}_i\|^2 + \frac{1}{2} \log \hat{\sigma}_i^2. \tag{7}$$

## A.3 Implementation details

In this section, we elaborate on the specifics of model implementation. In particular, we describe our methods of molecule digitization, fragment representation, encoder structure, and training schemes.

Table 6: Atom features

| Types of atoms | C, N, O, S, P, F, I, Cl, Br, * |
| --- | --- |
| GetDegree() | 0,1,2,3,4,5 |
| GetTotalNumHs() | 0,1,2,3,4 |
| GetImplicitValence() | 0,1,2,3,4,5 |
| GetIsAromatic() | Boolean |

**Molecular representation.** We present features used in representing atoms in Table 6, where '*' is a mark for the attachment site. We used RDkit [11] to extract the above features from given molecules. Node features of an atom are one-hot encoded and concatenated into a vector, which is then mapped through a trainable embedding layer to produce a dense vector.

**Fragment representation.** In order to create representations for fragments, we utilize the Morgan circular molecular fingerprint bit vector of size 1024 and radius 2 as implemented in RDKit with default invariants that use connectivity information similar to those used for the ECFP fingerprints. Morgan fingerprints are a commonly used rule-based representation for molecules.

**Encoder.** We used 3 layers of GCN [12] with ReLU activation between each layer. Node feature vectors are first mapped to a dense vector with a linear layer. A graph with dense node vectors $H^{(0)}$ and an adjacency matrix $A$ are then mapped through a GCN layer:

$$H^{(l+1)} = \text{AGG}(\text{ReLU}(\{\tilde{D}^{-1/2}\tilde{A}\tilde{D}^{-1/2}H^{(l)}W^{(l)}\})) \tag{8}$$

where $\tilde{A} = A + I$, $\tilde{D}_{ii} = \sum_j \tilde{A}_{ij}$. Graph embedding vector is acquired by readout operation on the final (the third) node embedding vectors, i.e. $H^{(3)}$. We only used sum operation for aggregation (AGG) and readout, considering graph isomorphism test in molecular graphs [13, 14].

**Training.** We have set a maximum number of actions in an episode as four in *de novo* generation and two in a scaffold-based generation, considering the distribution of the number of fragments in DUD-E compounds [15]. Figure 4 of [15] shows that the most popular number of fragments in DUD-E active compounds are around 4~6. Accordingly, we only considered four-action episodes in this work. We did not make our algorithm learn when to stop, but we leave this as our imminent future work.

With 64-dimensional embedding as default, we train our policy network, Q function network, graph encoder, $\alpha$ in SAC, and priority predictor using the Adam optimizer [16], an initial learning rate of 1e-3, weight decay of 1e-4, and a batch size of 256. Learning rates are reduced with ReduceLROnPlateau on PyTorch, with reduce factor 0.1 and patience steps 768 for policy and Q function, and 500 for priority predictor. In case of SAC, initial soft actor-critic $\alpha$ values are set to 1 and [min, max] value set to [0.05, 20]. The learning rate of $\alpha$ in soft actor-critic was initialized with 5e-4. Hyper-parameters for PER follow original setting in [8], with $\alpha$ in (3) set to 0.6, and $\beta$ in (4) set at $min(1.0, \beta_{init} + idx * (1.0 - \beta_{init})/\beta_{frames})$, where $\beta_{init} = 0.4$ and $\beta_{frames} = 1e + 5$. Every trainable parameter is initialized with Xavier uniform initializer [17].

**Random exploration** In order to encourage exploration, we let the model randomly generate the experience during the first 4,000 iterations. After the random exploration, the model generates experience by the policy. The random exploration has significantly improved the model performances.

**Computational resources.** We used Intel®Xeon®Silver4210 for CPU computation including docking score calculation, and NVIDIA TitanRTX for GPU computation. Each random seed was run on one GPU resource, where 5 random seeds shared one CPU resource for computation.

## A.4 Experimental settings

In this section, we elaborate on our experimental settings and procedures.

**Scoring function.** We use the docking program as a scoring function to compute the final state reward. While Autodock Vina [18] is the most popular docking program for virtual screening, we used QuickVina 2 instead of AutoDock Vina to increase the speed of docking computation since QuickVina 2 is known to provide 20.39-fold acceleration compared to AutoDock Vina. Also, the binding affinity predictions from QuickVina 2 and AutoDock Vina show Pearson's correlation coefficient of 0.911 [19]. Since 0.911 indicates a high correlation, we believe that using QuickVina 2 instead of AutoDock Vina would not seriously harm the fidelity.

The speed and accuracy of the docking simulation depend on its parameters – exhaustiveness in particular. While speed and accuracy are in a trade-off relationship, we choose a high speed, low accuracy setting (exhaustiveness = 1) to minimize the computational cost of model training.

Table 7: Docking configuration

|                      | Configuration |
| -------------------- | ------------- |
| exhaustiveness       | 1             |
| subprocess           | 10            |
| cpu per subprocess   | 1             |
| modes                | 10            |
| timeout (gen3d)      | 30 sec        |
| timeout (docking)    | 100 sec       |

We report the docking configuration we used for the experiments in Table 7. With our computational setting, docking calculation costs around 0.9 sec per sample.

**Protein targets.** Three protein targets **fa7 (FA7)**, **parp1 (PARP-1)**, **5ht1b (5-HT1B)** are chosen as design objectives to train the generative model on. While the model performance can greatly deviate according to its protein target, we carefully chose the targets to avoid bias in the experiments. The three targets have one of the highest AUROC scores when the protein-ligand binding affinities for DUD-E+ ligands are approximated with AutoDock Vina and the result was compared with ground truth, meaning that AutoDock Vina works fairly well for those three targets [20]. We assume that QuickVina 2, a derivative of AutoDock Vina, would similarly work well for the three targets. Also, three targets have different protein family memberships - fa7 (Coagulation factor VII) in protease family, parp1 (Poly [ADP-ribose] polymerase-1) in polymerase family, and 5ht1b (5-hydroxytryptamine receptor 1B) in G protein-coupled receptor family.

***De novo* and scaffold-based scenario.** *De novo* drug design is to generate molecules with high therapeutic potential from scratch. Scaffold-based drug design is to add or modify substructures from the given scaffold. In our experiments, we prepare a benzene ring as an initial molecule from *de novo* scenario since aromatic rings are common substructures in druglike molecules.

For scaffold-based scenarios, we detect scaffold for each target by the procedure described below and use those scaffolds as initial molecules. Note that our model requires the user to set the number of fragments to add to the initial molecule. We allow four augmentation steps for *de novo* designs and two augmentation steps for scaffold-based designs.

**Scaffold detection.** We detect scaffold of DUD-E [21] (fa7, parp1) or ChEMBL [22] (5ht1b) active compounds using MurckoScaffold function implemented by RDKit [11]. Then, we sort the scaffolds by frequency and take the most frequently observed scaffold as an initial molecule. We observed several active compounds which include chosen scaffold as their substructures, and carefully chose the attachment sites that connect the scaffold to surrounding atoms.

**Pharmacochemical filters.** We utilize widely-used pharmacochemical filters to assess the generated molecule's quality. We calculate the ratio of molecules accepted by each filter to total generated molecules and report the result in Section 4.2 Table 1. The filters are also used to exclude inappropriate fragments from the fragment library.

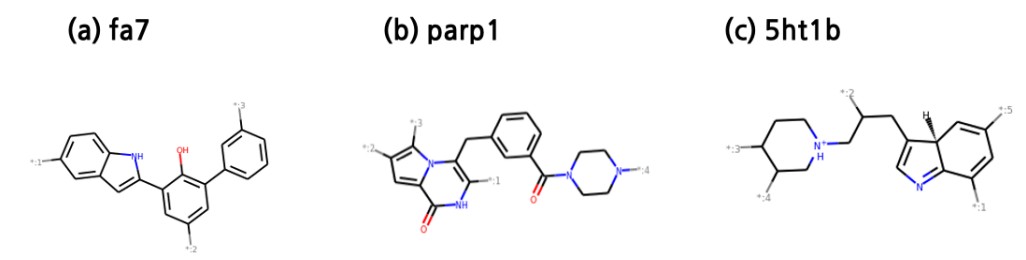

Figure 4: Detected scaffolds of fa7, parp1, and 5ht1b

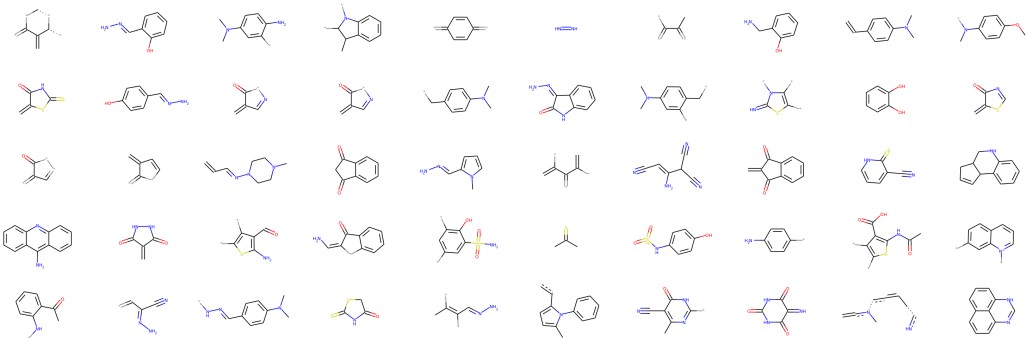

Figure 5: 50 examples of structural alerts in PAINS filter

**1) PAINS**: Pan-assay interference compounds (PAINS) [23] are chemical compounds that tend to bind nonspecifically with numerous biological targets rather than discriminately affecting one desired target, often giving false positive results in high-throughput screening [24, 25]. The PAINS filter contains 481 structural alerts.

**2) SureChEMBL Non MedChem-Friendly SMARTS**: SureChEMBL Non MedChem-Friendly SMARTS [26] is a set of structural alerts or toxicophores, which are substructures that are highly correlated with properties undesirable for drugs typically associated with human or environmental toxicity. When such structural alerts are used to filter medicinally unfriendly compounds in early-stage drug discovery, a significant reduction in compound failure rates in the clinic has been observed. The SureChEMBL Non MedChem-Friendly SMARTS contains 166 structural alerts [27].

**3) Glaxo Hard Filters**: Glaxo Hard Filters are a set of substructure filters that rejects compounds containing inappropriate functional groups, such as reactive functional groups, unsuitable leads (i.e., compounds which would not be initially followed up), and unsuitable natural products (i.e., derivatives of natural product compounds known to interfere with common assay procedures). The Glaxo Hard Filters contain 51 structural alerts. [28]

**Fragment library.** For **random fragments**, we fragmented 250k druglike molecules in the ZINC database [29] and filtered the fragments according to a number of atoms, radius, and frequency. We only took the fragments that contained fewer than 12 atoms, and we excluded fragments that appear only once or twice in the ZINC database.

In the fragment filtering procedure, fragments that might evoke RDKit parse errors were excluded. Also, if two fragments with the same graph structure have different attachment sites, we remove the one that has fewer attachment sites. For experiments in Section 4.3 and Section 4.4, we use **large fragment** library, which does not exclude filter-rejected fragments. We choose 91 fragments that appear most frequently in ZINC druglike molecules. For experiments in Section 4.2, we use **small library** where we excluded fragments rejected by PAINS, SureChEMBL, and Glaxo filters from **large library**. The **small library** consists of 66 fragments. The structures of the fragments are shown in Figure 8 and Figure 9. We provide Jupyter notebook for fragment library generation in our GitHub repository https://github.com/AITRICS/FREED.

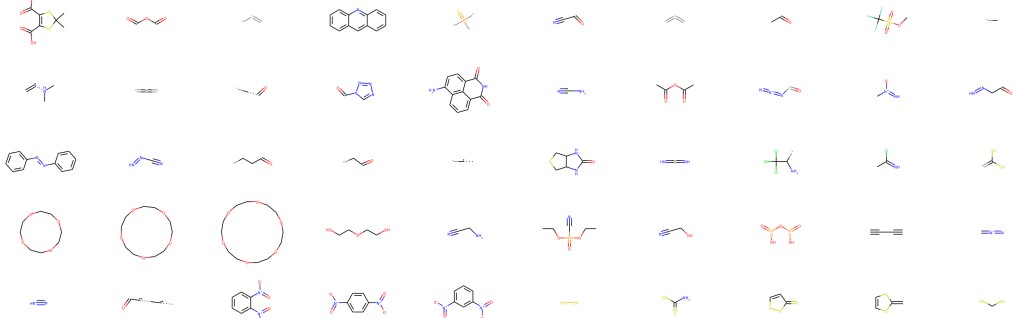

Figure 6: 50 examples of structural alerts in SureChEMBL Non MedChem-Friendly SMARTS

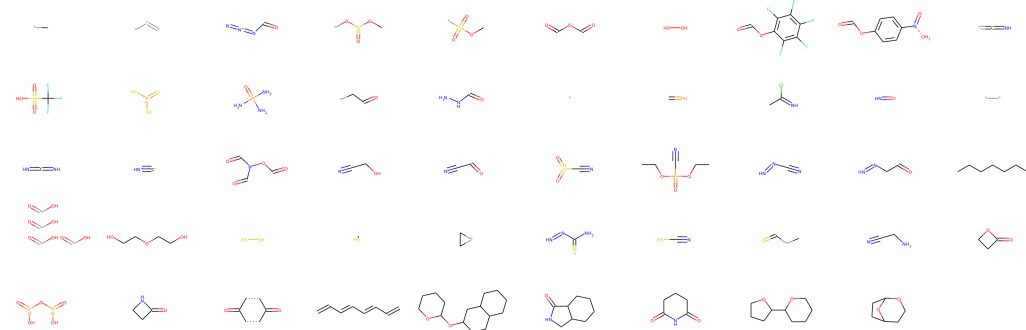

Figure 7: 49 examples of structural alerts in Glaxo Hard Filter

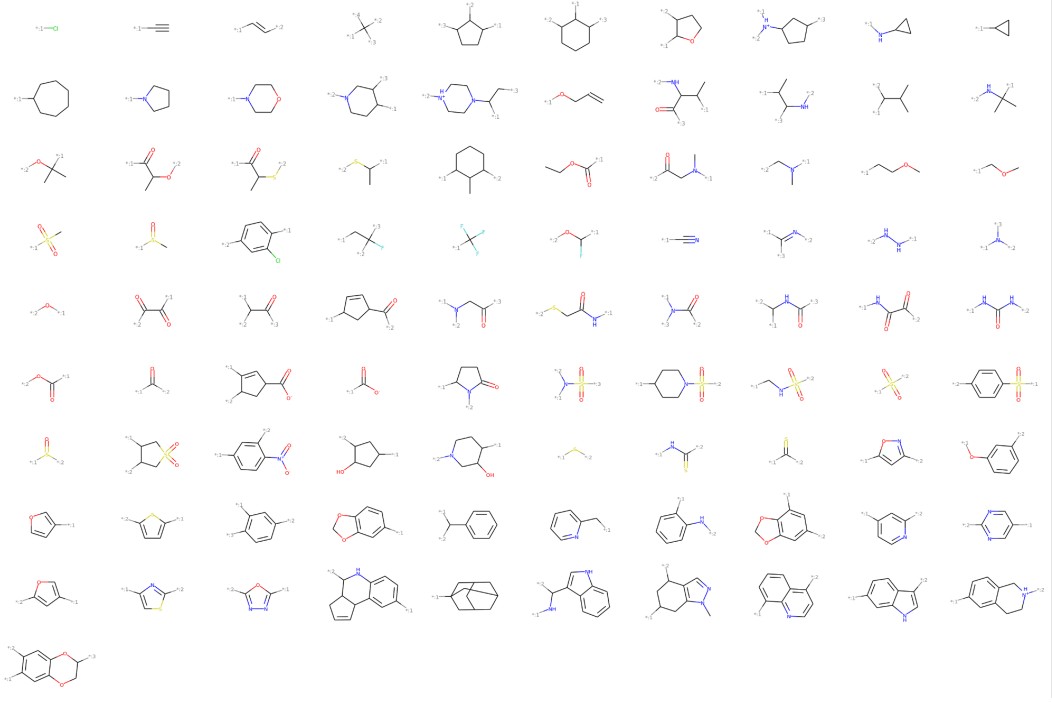

Figure 8: 91 fragments in **large library**.

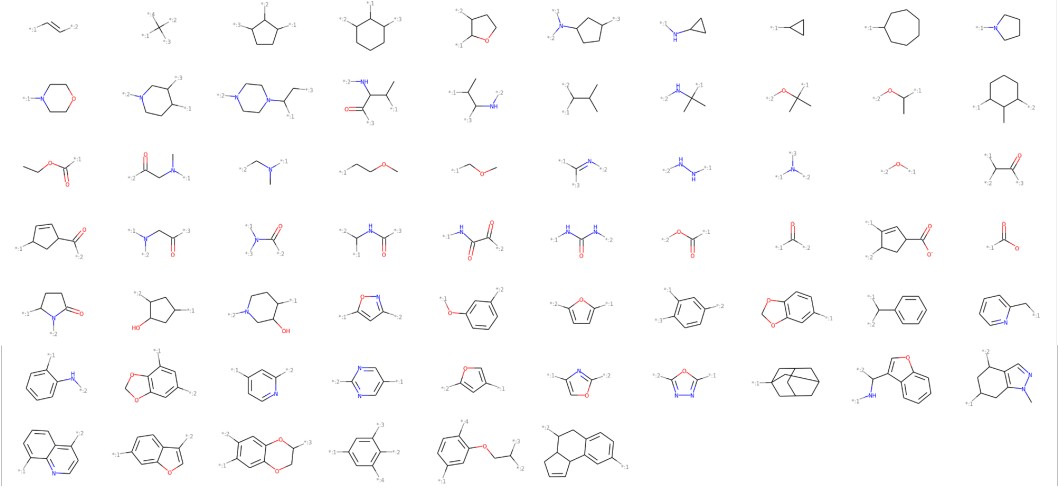

Figure 9: 66 fragments in **small library**.

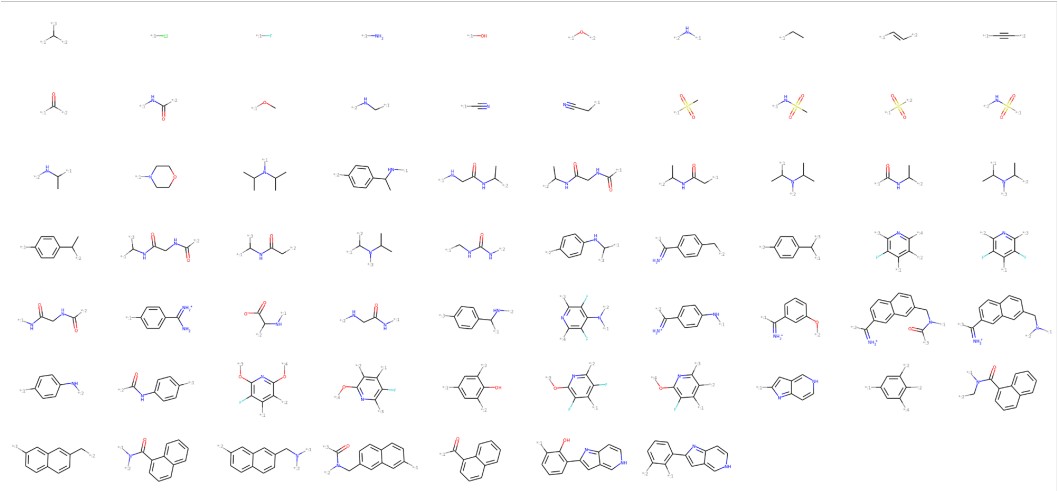

Figure 10: 67 fragments in **active library**.

## A.5   Baseline frameworks

We used MORLD and REINVENT as our baselines since those two models are utilized to optimize the docking score in previous works. While MORLD is an atom-based generative model, REINVENT can be considered as a representative SMILES-based generative model. We also compared our model with HierVAE. HierVAE is a strong variational autoencoder (VAE)-based method that generates molecular graphs using structural motifs as building blocks.

**MORLD.**   MORLD [30] is an atom-based molecular generative model guided by the MolDQN algorithm. We set a benzene ring as an initial molecule for MORLD training. We use QuickVina 2 docking program as a scoring function to train MORLD as in the original work. We jointly trained QED and SA scores as in the original work.

**REINVENT.**   REINVENT [31] is a smiles-based molecular generative model guided by an improved REINFORCE algorithm. We used the RNN model pretrained on a 1.5 million ZINC druglike molecule dataset. The pretrained model was provided by the original work [2]. While the Cieplinksi et

---

[2] https://github.com/MarcusOlivecrona/REINVENT

al. [32] and Bender et al. [33] utilized the SMINA docking program as its scoring function, we use QuickVina 2 docking program as a scoring function to reduce the computational cost.

**HierVAE.** We used the model checkpoint uploaded on the official GitHub repository, which was pretrained with ChEMBL molecules [3]. We also used the HierVAE vocab (fragment) library from the official GitHub. 89.6% fragments of the vocab are Glaxo-accepted molecules, 81.8% are SureChEMBL-accepted molecules, and 99.9% are PAINS-accepted molecules.

We then fine-tuned the model with respect to three protein targets (fa7, parp1, 5ht1b) using the corresponding DUD-E active molecules as the training set. After removing the molecules that evoke RDKit parse errors, we obtained as the training sets 99 samples for fa7, 433 samples for parp1, and 1129 samples for 5ht1b. We trained the models for the proteins until the loss converges. Since the 5ht1b training set has a larger size, the 5ht1b model converged more slowly than the other two, and we trained until 700 epochs for fa7 and parp1, and 800 epochs for 5ht1b. We generated 3,000 molecules with each model, and we computed the hit ratio and top 5% score.

For active learning scheme (HierVAE(AL)), we first fine-tuned the ChEMBL pretrained HierVAE with the active molecule set from DUD-E. Then, we generated 3,100 molecules from the fine-tuned HierVAE, computed the molecules' docking scores, and gathered the set of molecules whose docking scores are better than the active threshold (=the median of the docking scores of the active molecules), and fine-tuned the model with these molecules. We repeated the collecting of the molecules and fine-tuning twice. In this scheme, we perform ~6,000 docking computations, which is roughly twice of the other methods. The number of selected molecules in the first round of training were 1,138 for 5ht1b, 1,732 for parp1, and 291 for fa7. For second round, 585 molecules for 5ht1b, 1,790 for parp1, and 346 for fa7 were used for training. The model performance was improved in terms of both hit ratio and top 5% score, compared to the one-time training scheme.