# OpenReview forum: "Hit and Lead Discovery with Explorative RL and Fragment-based Molecule Generation"
_NeurIPS.cc/2021/Conference — NeurIPS 2021 Poster_

### Official Review · Reviewer_bYsg · 2021-07-06

**Rating:** 4
**Confidence:** 3

**Summary:**

This paper is concerned about optimizing molecules by assembling fragments using reinforcement learning. The main idea to generate "drug-like" molecules is to generate a molecule by combining chemically realistic and pharmacochemically acceptable fragments, while restricting the attachment sites of the fragments. This generation process coupled with the explorative RL algorithm is the proposed method.

The effectiveness of the proposed method is investigated by comparing it with two existing methods. The experimental results suggest that (i) the molecules generated by the proposed method are more likely to pass the pharmacochemical filters than the existing methods, (ii) the proposed method achieves higher or comparable hit ratio and top 5% scores than the existing methods, (iii) the explorative RL algorithm used in the proposed method is suitable to this task than the other RL methods.

The authors also provide a case study on drug design in both de novo scenario and scaffold-based one.

**Limitations And Societal Impact:**

The fragment-based generation could be limited as compared to the atom-wise generation in that the generated molecules are less diverse; this is because the generated molecules are restricted to the fragment library. So, I would recommend the authors to investigate the diversity of generated molecules.

**Main Review:**

# Summary
- I like the authors work on realistic problems rather than synthetic ones such as logP optimization, formally considered in the literature.
- Technically, the proposed method seems to be minor contributions to the literature. For example, the fragment-based generation with connectivity preservation has been introduced in the paper "Molecular Hypergraph Grammar with Its Application to Molecular Optimization" by Kajino (ICML-19), and a similar generation process is coupled with RL in the paper "Reinforced Molecular Optimization with Neighborhood-Controlled Grammars" by Xu et al. (NeurIPS-20).
- There are two major concerns; see below.

# On the hypothesis
An important hypothesis of the present work is that attaching a chemically realistic and pharmacochemically acceptable fragment units leads to realistic and acceptable molecules. The filters used in the experiments seem to be based on the same hypothesis, but I am curious whether this hypothesis is widely accepted or not. If this hypothesis is just a necessary condition for a molecule to be acceptable, it should be noted as a limitation of this work.

# Comparison to the existing methods
In order to validate the proposed method, it is necessary to empirically compare the proposed method with the existing methods, at least those mentioned below.
- In l.93-95, the authors state that their connectivity-preserving fragmentation and augmentation procedure is beneficial, but this is not obvious and should be empirically confirmed.
- In l.99-107, the authors state that the forward synthesis methods will not be able to handle docking score optimization, but this is also not obvious and should be empirically confirmed.

**Time Spent Reviewing:**

4 hours

---

> ### Author Response · Authors · 2021-08-10
> **Response**
>
> We sincerely appreciate your constructive and helpful comments. We initially address all your comments below:
>
> **Q1.** On the hypothesis: An important hypothesis of the present work is that attaching a chemically realistic and pharmacochemically acceptable fragment unit leads to realistic and acceptable molecules. The filters used in the experiments seem to be based on the same hypothesis, but I am curious whether this hypothesis is widely accepted or not. If this hypothesis is just a necessary condition for a molecule to be acceptable, it should be noted as a limitation of this work.
>
> **A1.** The three medicinal chemistry filters are widely used for drug discovery. Those filters are developed by global pharmaceutical companies such as Novartis and Glaxo. Although passing the three filters does not guarantee the 100% acceptability of the molecules as drugs, our work functions as a proof of concept that our model is able to plug in any expert knowledge related to substructural alerts by controlling the fragment library. The limitation is that we do not know yet the perfect sufficient condition for a drug candidate to be pharmacochemically acceptable, but this is a limitation that can be overcome from the advances in medicinal chemistry, not from our model design.
>
> * * *
>
> **Q2.** In order to validate the proposed method, it is necessary to empirically compare the proposed method with the existing methods, at least those mentioned below.
>
> **Q2-1.** In l.93-95, the authors state that their connectivity-preserving fragmentation and augmentation procedure is beneficial, but this is not obvious and should be empirically confirmed.
>
> **A2-1.** This assumption can be easily verified with domain knowledge. Without the connectivity-preserving fragmentation and augmentation, uncommon and often chemically unrealistic molecules will be generated frequently. For example, while a phenyl ring with two substituent groups is conventional, a phenyl ring with four substituent groups is an uncommon structure and even unstable due to steric strains. In this case, our connectivity-preserving fragmentation and augmentation procedure enables the model to only consider the phenyl rings that have two substituent groups.
>
> **Q2-2.** In l.99-107, the authors state that the forward synthesis methods will not be able to handle docking score optimization, but this is also not obvious and should be empirically confirmed.
>
> **A2-2.** We agree that our conjecture on the forward synthesis methods is not empirically validated. However, the forward synthesis method we mentioned in the manuscript [1] hasn’t released the code of their model yet. We would be happy to test our hypothesis empirically when the code is released.
>
> * * *
> **Q3.** The fragment-based generation could be limited as compared to the atom-wise generation in that the generated molecules are less diverse; this is because the generated molecules are restricted to the fragment library. So, I would recommend the authors to investigate the diversity of generated molecules.
>
> **A3.** Thank you for the suggestion. We computed the Internal Diversity ($\text{IntDiv}_1$) of our model and the baseline models according to the widely used MOSES benchmark [2]. Observing the result below, despite the strong constraint on the molecule structure, our model shows moderate diversity. Notably, our model trained with the larger (number of fragments = 91) library shows better diversity and uniqueness compared to our model trained with the smaller (number of fragments = 66) library. This result shows the effect of the size of the fragment library, and from this observation, we suggest using sufficiently large libraries.
>
> ours (small): FREED(PE) trained with the small library used for Section 4.2.
> ours (large): FREED(PE) trained with the large library used for Section 4.3.
>
> **fa7**
>
> |  | Internal diversity | uniqueness |
> | --- | --- | --- |
> | ours (small) | 0.778 (0.021) | 0.723 (0.135) |
> | ours (large) | 0.837 (0.014) | 0.914 (0.162) |
> | REINVENT | 0.862 (0.008) | 0.988 (0.008) |
> | MORLD | 0.897 (0.009) |1.000 (0.000)  |
>
> **parp1**
>
> |  | Internal diversity | uniqueness |
> | --- | --- | --- |
> | ours (small) | 0.787 (0.015) | 0.557 (0.141) |
> | ours (large) | 0.817 (0.018) | 0.827 (0.160)|
> | REINVENT | 0.860 (0.007) | 0.993 (0.004) |
> | MORLD | 0.895 (0.001) | 1.000 (0.001) |
>
> **5ht1b**
>
> |  | Internal diversity | uniqueness |
> | --- | --- | --- |
> | ours (small) | 0.782 (0.017) | 0.592 (0.243) |
> | ours (large) | 0.817 (0.010) | 0.801 (0.191) |
> | REINVENT | 0.859 (0.009) | 0.982 (0.010) |
> | MORLD | 0.898 (0.001) | 1.000 (0.000) |
>
>
> * * *
> [1] Gottipati, Sai, et al., "Learning To Navigate The Synthetically Accessible Chemical Space Using Reinforcement Learning", arXiv:2004.12485 [cs.LG]
>
> [2] Polykovskiy, Daniil, et al., "Molecular Sets (MOSES): A Benchmarking Platform for Molecular Generation Models", Frontiers in Pharmacology, 2020

---

> > ### Comment · Reviewer_bYsg · 2021-08-14
> > **Re: Authors' rebuttal**
> >
> > I appreciate the authors spend much time to answer my questions.
> >
> > ## On A1.
> > I understand this hypothesis is common in the pharma industry and that it is just one proxy to distinguish realistic molecules and the others. Then, I would recommend the authors to pay their great attention to clarify it in the main text; otherwise, readers could misunderstand that the hypothesis is true.
> >
> > ## On A2-1.
> > I do understand the authors' expectation of the benefit of a model with the connectivity-preserving property. However, a model without the property may be able to avoid from unstable structures if it learns the mechanism from data (or in other words, we cannot prove that a model without the property cannot avoid from unstable structures). Therefore, I consider this statement can be confirmed only by empirical studies.
> >
> > ## On A2-2.
> > In my opinion, no public code cannot be an excuse for no performance comparison, considering the potential difficulty to implement the method by Gottipati et al. At least, statements in l.99-107 should be confirmed with some scientific basis or otherwise should be removed.
> >
> > ## On A3.
> > I appreciate the authors to investigate the internal diversity. This result is very informative for readers to understand the properties of both the proposed and existing methods, and I would recommend the authors to present it in the paper.
> >
> >
> > Overall, while I like the authors working on a realistic problem setting, writing is not very clear and sometimes lacks scientific evidence, which strongly discourages me to vote for acceptance. I would recommend the authors to revise and resubmit the paper so that most of important statements are supported by some scientific evidence.

---

### Official Review · Reviewer_ADzq · 2021-07-11

**Rating:** 4
**Confidence:** 4

**Summary:**

The paper presents a reinforcement learning approach in which molecule fragmetns are combined together in order to construct realistic and pharmacologically relevant molecules. The approach relies on considerable use of domain knowledge, since the library of fragments that are used to generate the molecules are themselves realistic and pharmacologically relevant and they do not exhibit undesired properties, e.g. toxicity or low reactivity. The overall goal of the method presented in the paper is to learn to generate molecules that will dock to given proteins. With respect to the RL problem the reward that is used is based on the docking score as well as on a pharmacochemical filter scores. This is a hard-exploration learning problem due to very sparse rewards, which according to my understanding are available once all the molecule has been generated. To address the hard-exploration problem the paper relies on a number of variants of SAC which make use of prioritise replay. The paper compares against a couple of baselines with favorable results. My impression is that the main advantage of the method with respect to the baselines is that the library of fragments over which it is operating is carefully selected towards producing molecules that are pharmacoligically relevant and acceptable, and less the algorithmic contributions, e.g prioritised replay.



**Limitations And Societal Impact:**

yes they have.

**Main Review:**

While the paper is in  general easy to ready, I had issues on a number of occasions. For example the I found the discussion of the prioritised reaply methods that the paper used confusing and unclear. I believe I also missed a precise definition of the reward that it was used, in the appendix it is described as "docking score reward + pharmacochemical filter reward" but what are the filters that are used?

In the experiments the paper compares against two baselines, for which not many details are given except the fact that they are generative models, though in the description in the introduction it seems that they are rather reinforcement learning algorithms. I could not help but wonder whether a quantitative comparison with proper generative models, many of which the paper discusses, would not be possible? Many of these generative models have excellent performance for the tasks on which they have been used/benchmarked (and they are also readily available for conditional generation), though I am not sure any of them has been tried on the docking generation problem. If we were to take a conditional generation approach on the docking docking problem I guess the task would be something like given a particular target learn the distribution of the molecules that dock to it. I am not sure whether this is easy to do, since there would be issues even with the generation of proper training sets. In any case I am not sure that one can so easily dismiss these models, and with respect to that aspect I have the feeling that the set of baselines is not complete, though I admit benchmarking these methods on the docking problem is not obvious.


lines 96-98: Is there evidence out there that learning to reconstruct might harm diversity? Why having a decoder that is able to reconstruct will have less chances to produce diverse molecules? and by the way the same argument can also be given for the method presented in this paper, i.e. isnt't the fact that the approach is based on fragments limit the diversity of the generated molecules, since these can only be combinations of existing fragments? which in fact seems to be supported by the experiments.

line 145: I am not sure how should I interpret the "predict a next state"? should it be simply action prediction?

Actions:
* lines 154-155: What is the action space? The first action selects the docking point in the so far generated molecule, the possibilities can be very different depending on the so far generated molecule, how this is handled? or in other words what is the structure of the output of $\pi_{\text{act1}}$? similar question for second and third action selection.
Relate to this, in the appendix, line 86 states that the maximum number of actions in an episode is four? does that mean that there only four transitions, i.e. all episodes have a length of four? is there no possibility that after fours steps we might end up with an incomplete molecule?

line 169: for my undestanding, all rewards are zero except the last one which is given by the docking score and the pharmacological filter rewards?

lines 173-174: eq 5: Isn't the expectation missing the $\pi(a|s)$ term ? Also the expectation is approximated by the sample based estimate but it is not equal to it.
To avoid ambigiuty would be useful to have the two sums, the one that comes from the expecation and the one over the actions for the entropy.


Hard exploration part:
* lines 187-193: To address the hard-exploration the paper couples SAC with prioritized replay where the priority of a state is either the predictive error of a reward prediction network
 or the uncertainty of that reward as estimated by a neural net with dropout. Since the priority is based on reward uncertainty shouldn't this be computed based on state-action
pairs, instead of states? I am not sure I understand what is the reward of a state. By the way the auxiliary network that estimates reward is independent of the critic of SAC? In the appendix
lines 53-61 (curiosity driven learning) there is a discussion of reward prediction, though only the loss is given, $\hat{y_i}$ is the estimated reward of some given state? and the true reward
of that state will be almost always 0? unless the molecule is complete, in which case it will be the docking score plus the pharma filters?

* lines 194-199: I have to say that here I got confused, I am not sure I see how 1) and 2) relate do they provide two different ways to prioritise experience replay? one seems to focus on states only,
while in 2) the focus seems to be transitions? By the way a transition is the quadruple (s,a,r,s')? and is the priority of a given transition based on the reward prediction or on the docking score estimation? both are mentioned in the paragraph, and I thought the docking score is only given for complete molecules but the paragraph makes reference to docking score estimation for a given state.

Experiments:
* baselines: a couple of sentences that describe the type of generative approach that REINVENT and MORLD follow would have been appreciated.

**Time Spent Reviewing:**

7

---

> ### Author Response · Authors · 2021-08-10
> **Response (2/2)**
>
> **Q10.** line 169: For my understanding, all rewards are zero except the last one which is given by the docking score and the pharmacological filter rewards?
>
> **A10.** Yes, all external rewards are zero except for the last state at the end of the episode.
> * * *
> **Q11**. lines 173-174: eq 5:
> (1) Isn't the expectation missing the π(a|s) term?
> (2) Also, the expectation is approximated by the sample based estimate but it is not equal to it. To avoid ambiguity it would be useful to have the two sums, the one that comes from the expectation and the one over the actions for the entropy.
>
> **A11.**
> (1) No, It can be expressed as $E_{s_t \sim D}$ $[\alpha\log{\pi(a|s)}]$, or either $\sum[{\pi(a|s)}\alpha\log{\pi(a|s)}]$.
> (2) We will change the summation term into $\sum_{D}$, to express that the term inside the summation is summed over samples that are sampled from the replay buffer. Since the term inside the summation is only summed in terms of samples that are from replay buffer, using one summation term is enough.
>
> * * *
> **Q12.** lines 187-193:
> (1) Since the priority is based on reward uncertainty shouldn't this be computed based on state-action pairs, instead of states? I am not sure I understand what is the reward of a state.
> (2) By the way the auxiliary network that estimates reward is independent of the critic of SAC? In the appendix lines 53-61 (curiosity driven learning) there is a discussion of reward prediction, though only the loss is given, yi^ is the estimated reward of some given state? and the true reward of that state will be almost always 0? unless the molecule is complete, in which case it will be the docking score plus the pharma filters?
>
> **A12.**
> (1) In our work, priority is computed based on state-space instead of state-action space, which we refer to (Thiede et al. 2020). The reward of the state (molecule) is computed from the docking program which accepts molecules (states) as input.
> (2) We apologize for the confusion. We will clarify this point in the revision. Yes, the docking score is only given for complete molecules. However, in cases of incomplete molecules (the states in the middle of the episodes) that we did not compute docking score, we used Q function prediction of the states as a substitute for the actual docking score.
>
> * * *
> **Q13.** lines 194-199:
> (1) I am not sure I see how 1) and 2) relate do they provide two different ways to prioritise experience replay? one seems to focus on states only, while in 2) the focus seems to be transitions?
> (2) By the way a transition is the quadruple (s,a,r,s')? and is the priority of a given transition based on the reward prediction or on the docking score estimation?
> (3) Both are mentioned in the paragraph, and I thought the docking score is only given for complete molecules but the paragraph makes reference to docking score estimation for a given state.
>
> **A13.**
> (1) 1) proposes priority estimation using predictive error or Bayesian uncertainty. 2) describes how both estimators are structured and function in detail.
> (2) Yes, a transition is the quadruple (s, a, r, s'). Also, for PER(PE), the priority of a given transition is based on the difference between the reward prediction and the docking score computation.
> (3) We apologize for the confusion. We will clarify this point in the revision. As we explained in **A12**, in cases of incomplete molecules (the states in the middle of the episodes) that we did not compute docking score, we used Q function prediction of the states as a substitute for the actual docking score.
>
> * * *
> **Q14.** baselines: a couple of sentences that describe the type of generative approach that REINVENT and MORLD follow would have been appreciated.
>
> **A14.** Thank you for the suggestion. we will add the description in the revision.
> * * *

---

> ### Author Response · Authors · 2021-08-10
> **Response (1/2)**
>
> We sincerely appreciate your constructive and helpful comments. We initially address all your comments below:
>
> **Q1.** I found the discussion of the prioritised replay methods that the paper used confusing and unclear. I believe I also missed a precise definition of the reward that was used, in the appendix it is described as "docking score reward + pharmacochemical filter reward" but what are the filters that are used?
>
> **A1.** The information of the filters is already described in Section 4.1 and Appendix A.4. We apologize for the confusion, and we will revise the manuscript for better clarity.
> * * *
> **Q2.** In the experiments the paper compares against two baselines, for which not many details are given except the fact that they are generative models, though in the description in the introduction it seems that they are rather reinforcement learning algorithms.
>
> **A2.** Thank you for pointing this out. We will add the description in the revision.
> * * *
> **Q3.** I could not help but wonder whether a quantitative comparison with proper generative models, many of which the paper discusses, would not be possible? Many of these generative models have excellent performance for the tasks on which they have been used/benchmarked (and they are also readily available for conditional generation), though I am not sure any of them has been tried on the docking generation problem.
>
> **A3.** We compared our model with REINVENT and MORLD because those were the two models that were previously tested on a docking score optimization task. These two models are considered as strong generative models and were cited many times (429 times for REINVENT, 4 times for MORLD). We additionally performed a benchmark test on HierVAE, which is one of the most popular recent generative models. We report the result below as an answer to **Q4**.
> * * *
> **Q4**. If we were to take a conditional generation approach on the docking problem I guess the task would be something like given a particular target learn the distribution of the molecules that dock to it. I am not sure that one can so easily dismiss these models, and with respect to that aspect, I have the feeling that the set of baselines is not complete, though I admit benchmarking these methods on the docking problem is not obvious.
>
> **A4.** Thank you for the helpful suggestion. We added HierVAE [2], a strong generative model introduced in 2020, as our additional baseline model. As a result, our model FREED(PE) still outperformed HierVAE in terms of hit ratio and top 5% score.
>
> We used the model checkpoint uploaded in the official GitHub, which was pre-trained with ChEMBL molecules. We then fine-tuned the model with respect to three protein targets (fa7, parp1, 5ht1b) using the corresponding DUD-E active molecules as the training set. After removing the molecules that evoke RDKit parse errors, we obtained as the training sets 99 samples for fa7, 433 samples for parp1, and 1129 samples for 5ht1b. We trained the models for the proteins until the loss converges. Since the 5ht1b training set has a larger size, the 5ht1b model converged more slowly than the other two, and we trained until 700 epochs for fa7 and parp1, and 800 epochs for 5ht1b. We generated 3,000 molecules with each model, and we computed the hit ratio and top 5% score as follows:
>
> **Target: fa7**
>
> | Performance scores | hit ratio [%] | top 5% score |
> | --- | --- | --- |
> | HierVAE | 2.247 (0.243) | -9.343 (0.228) |
> | ours (small) | 28.353 (18.033) | -10.906 (0.425) |
> | ours (large) | 26.360 (6.534) | -10.429 (0.483) |
>
> **Target: parp1**
>
> | Performance scores | hit ratio | top 5% score |
> | --- | --- | --- |
> | HierVAE | 12.093 (0.652) | -11.932 (0.399) |
> | ours (small) | 33.544 (11.766) | -13.772 (0.542) |
> | ours (large) | 46.447 (8.938) | -13.548 (0.561) |
>
> **Target: 5ht1b**
>
> | Performance scores | hit ratio | top 5% score |
> | --- | --- | --- |
> | HierVAE | 8.535 (0.347) | -11.572 (0.509) |
> | ours (small) | 46.266 (5.648) | -13.527 (0.643) |
> | ours (large) | 49.360 (5.806) | -13.297 (0.588) |
>
> 'ours (small)' indicates FREED(PE) trained with the small fragment library that was used for Section 4.2 experiments. 'ours (large)' indicates FREED(PE) trained with the large library that was used for Section 4.3 experiments. As expected our FREED largely outperforms HierVAE. This experimental comparison against HierVAE will be a valuable addition to further show the effectiveness of our drug discovery framework, and we sincerely thank you for the helpful suggestion.
>
> * * *
> **Q5.** lines 96-98: Is there evidence out there that learning to reconstruct might harm diversity? Why having a decoder that is able to reconstruct will have less chances to produce diverse molecules?
>
> **Q6.** And by the way the same argument can also be given for the method presented in this paper, i.e. isn't the fact that the approach is based on fragments limit the diversity of the generated molecules, since these can only be combinations of existing fragments? which in fact seems to be supported by the experiments.
>
> **A5 & A6.**
> In theory, it is true that fragment-based molecule generation may limit the diversity of generated molecules. However, our model grasps chemical-realisticness through fragment-based generation, while we solve the problem of diversity with explorative experience replay which we also proposed. **Balancing required constraints (realistic molecules) and diversity(exploration) is the main contribution of our paper.**
>
> We computed the Internal Diversity ($\text{IntDiv}_1$) of our model and the baseline models according to the widely used MOSES benchmark [1]. Observing the result below, **despite the strong constraint on the molecule structure, our model shows moderate diversity**. The trade-off between required constraints and diversity is also observed from comparing ours (small) and ours (large).
>
> To address **Q5**, we also compare the diversity with HierVAE. While $\text{IntDiv}_1$ is in fact moderate for HierVAE, uniqueness highly deteriorated. Also, $\text{IntDiv}_1$ seems to be affected by the size of the training set; fa7-HierVAE's $\text{IntDiv}_1$ is quite low, probably because of the small size of the training set (n=99). Considering that in many cases there are very few known active molecules for a given target, the diversity of molecules generated from distributional learning might get easily deteriorated.
>
> ours (small): FREED(PE) trained with the small library used for Section 4.2.
>
> ours (large): FREED(PE) trained with the large library used for Section 4.3.
>
> **fa7**
>
> |  | Internal diversity | uniqueness |
> | --- | --- | --- |
> | ours (small) | 0.778 (0.021) | 0.723 (0.135) |
> | ours (large) | 0.837 (0.014) | 0.914 (0.162) |
> | REINVENT | 0.862 (0.008) | 0.988 (0.008) |
> | MORLD | 0.897 (0.009) |1.000 (0.000)  |
> | HierVAE | 0.746 (0.006) | 0.138 (0.006) |
>
> **parp1**
>
> |  | Internal diversity | uniqueness |
> | --- | --- | --- |
> | ours (small) | 0.787 (0.015) | 0.557 (0.141) |
> | ours (large) | 0.817 (0.018) | 0.827 (0.160)|
> | REINVENT | 0.860 (0.007) | 0.993 (0.004) |
> | MORLD | 0.895 (0.001) | 1.000 (0.001) |
> | HierVAE | 0.838 (0.003) | 0.350 (0.017) |
>
> **5ht1b**
>
> |  | Internal diversity | uniqueness |
> | --- | --- | --- |
> | ours (small) | 0.782 (0.017) | 0.592 (0.243) |
> | ours (large) | 0.817 (0.010) | 0.801 (0.191) |
> | REINVENT | 0.859 (0.009) | 0.982 (0.010) |
> | MORLD | 0.898 (0.001) | 1.000 (0.000) |
> | HierVAE | 0.839 (0.003) | 0.214 (0.006) |
>
> * * *
> **Q7.** line 145: I am not sure how I should interpret the "predict a next state"? should it be simply action prediction?
>
> **A7.** Thank you for pointing out the confusion, and we will clarify this point in the revision. We meant “predict the actions”.
> * * *
> **Q8.** lines 154-155: What is the action space? The first action selects the docking point in the so far generated molecule, the possibilities can be very different depending on the so far generated molecule, how this is handled? or in other words what is the structure of the output of πact1?
>
> **A8.** The structure of the output of $\pi_{act1}$ is given as the chosen index among attachment points. For training, the probability of $\pi_{act1}$ is padded to a certain dimensional (e.g.128 dimension) vector to be used in updating parameters.
> * * *
> **Q9.** Similar question for second and third action selection. Relate to this, in the appendix, line 86 states that the maximum number of actions in an episode is four? does that mean that there only four transitions, i.e. all episodes have a length of four? is there no possibility that after fours steps we might end up with an incomplete molecule?
>
> **A9.** According to Figure 4 of [3], the most popular number of fragments in DUD-E active compounds are around 4~6. Considering that, we only considered four-action episodes in this work. We didn’t make our algorithm learn when to stop, but we leave this as our imminent future work.
> * * *
>
> [1] Polykovskiy, Daniil, et al., "Molecular Sets (MOSES): A Benchmarking Platform for Molecular Generation Models", Frontiers in Pharmacology, 2020
>
> [2] Jin, Wengong, et al., "Hierarchical Generation of Molecular Graphs using Structural Motifs", arXiv:2002.03230
>
> [3] Naderi, Misagh, et al., "A graph-based approach to construct target-focused libraries for virtual screening", J Cheminform. 2016 Mar 15;8:14

---

> > ### Comment · Reviewer_ADzq · 2021-09-01
> > **Acknowledging authors responce**
> >
> > I would like to thank the authors for their answers. I also appreciate the fact that they included a generative model in their baselines. Unfortunately, even though the additions and clarifications improve the paper, I still believe that it does not meet the NeurIPS bar mostly due to what I believe is a somehow limited novelty from a modeling point of view.
> >
> > For future versions I have a couple of comments, considering the hiererchical VAE, was this use in a conditional generative setting, where the model was trained given as a condition the desired protein and then would learn to generate appropriate compounds? or was it used as a generative model, where one generative model was produced for each protein? I believe it is the latter from what I understood from the authors responce. The result of that is a very small training set for each protein and the impossibility to exploit commonalities between the different proteins. Ideally a conditional generation model should be trained jointly over different proteins on which we want to dock. Then on test time we would challenge the model with potentially a new protein and it should output compounds that dock to it.
> >
> > I also believe that the fixed number of actions is a rather important limitation, in particular because it might produce ?unfinished? compounds.
> >
> > Again I would like to thank the authors for their response.

---

### Official Review · Reviewer_TmJ6 · 2021-07-16

**Rating:** 5
**Confidence:** 3

**Summary:**

The paper proposes a RL framework to generate molecules by assembling molecular fragments, such that the ending result has optimized docking scores. The RL model uses docking score as a reward function; an autoregressive policy network which chooses among three possible actions to modify an existing molecule (source attachment site, fragment to attach, target attachment site); a Soft Actor-Critic algorithm to learn the optimal policy, and explores different Prioritized Experience Replay strategies to produce diverse molecules. In the experiments, the authors show that the model can produce molecules with enhanced docking scores for some reference proteins, both in a de novo scenario and scaffold-based scenario.

**Ethical Concerns:**

No ethical concerns.

**Limitations And Societal Impact:**

Limitations are discussed to some extent. One that I find critical for these types of models is computational time. It takes a long time to compute accurate docking simulations. In this paper, the authors use a docking program that is known to produce faster but less reliable docking estimations. However, no one would use this program in a practical scenario; hence, it should be properly discussed by the authors what is the computational burden to train the model for its usage in the real world, not just for experimenting.

**Main Review:**

The main idea behind the paper is not novel (using docking scores as a reward function), but the proposed solution is interesting in principle. However, the paper is not written well; there are inconsistencies in notation (e.g. h_g is used for a graph representation, but then ECFP(h_g_{cand}) is used, while ECFP should take as input a molecule, not a representation), confusingly defined symbols (e.g. z_{1st}^{act1}), undefined acronyms (TD, BU, PPO), misspelled acronyms (BE instead of BU), which made reading this paper very difficult. Furthermore, the English language is really faltering sometimes. First and foremost, I suggest that the paper undergoes serious proofreading, otherwise I will be forced to confirm my (temporary) rejection.

On the technical side, I have several concerns:
- on page 6, the authors affirm that the model has been trained on 3 different proteins. However, the results reported in Table 1 concern only one of them (fa7). I checked the supplementary material (not very carefully, I reckon) and there is no sign of the evaluation on those proteins. Why is that so?
- on Figure 7b)  (and in Figure 7a) for protein 5htb1) the results basically say that the proposed method is no more effective than attaching fragments randomly to generate higher docking molecules, which makes me wonder if using PER strategies is effective at all;
- how are bond types handled by the GCN? Judging from line 150, the GCN doesn't handle edge types, which is quite unlikely.
- What are the novelty rates (i.e. the ratio of generated molecules that are not in the training set) of the model? Why you didn't report them?
- Similarly, the authors use several techniques to improve the diversity when generating molecules. So I would have expected that the chemical diversity of the 3000 sample was measured, when in fact it wasn't.
- When does the generative process stop? I couldn't see it explained on the paper. Perhaps I can find it in the supplementary material? It's information that I would like to read in the paper itself, though.

__Overall judgement__

Right now, I recommend rejection, given the technical doubts but most of all given the poor writing quality. I am open to revising my score, if the authors will put effort into making this paper readable and comprehensible (in which case, I will also go through the supplementary material more carefully).

**Time Spent Reviewing:**

4

---

> ### Author Response · Authors · 2021-08-09
> **Response**
>
> We sincerely appreciate your constructive and helpful comments.
>
> **Q1.** The paper is not written well. U, PPO). Furthermore, the English language is really faltering sometimes. First and foremost, I suggest that the paper undergoes serious proofreading, otherwise I will be forced to confirm my (temporary) rejection.
>
> **A1.**  We apologize for the insufficient readability of our text. We have gone through all the possible grammatical errors and notations, and we have greatly increased the quality of the English grammar and writing.
>
> * * *
>
> **Q2.** On the technical side, I have several concerns:
> On page 6, the authors affirm that the model has been trained on 3 different proteins. However, the results reported in Table 1 concern only one of them (fa7). I checked the supplementary material (not very carefully, I reckon) and there is no sign of the evaluation on those proteins. Why is that so?
>
> **A2.** Since we had to train our model on a fragment library different from the experiments in Section 4.3, we only reported quality scores of fa7 due to the limitations in computational resources.
>
> We here additionally report the quality scores of FREED(PE) using the small and large fragment library here.
>
> **The filtered small library (number of fragments = 66)**
>
> | Quality scores | Glaxo | SureChEMBL | PAINS | validity | uniqueness |
> | --- | --- | --- | --- | --- | --- |
> | fa7 | 0.996 (0.001) | 0.808 (0.049) | 0.991 (0.002) | 1.000 (0.000) | 0.723 (0.135) |
> | parp1 | 0.995 (0.002) | 0.854 (0.050) | 0.991 (0.010) | 1.000 (0.000) | 0.557 (0.141) |
> | 5ht1b | 0.995 (0.005) | 0.823 (0.106) | 0.990 (0.007) | 1.000 (0.000) | 0.592 (0.243) |
>
> **The unfiltered large library (number of fragments = 91)**
>
> | Quality scores | Glaxo | SureChEMBL | PAINS | validity | uniqueness |
> | --- | --- | --- | --- | --- | --- |
> | fa7 | 0.754 (0.072) | 0.558 (0.094) | 0.623 (0.201) | 1.000 (0.000) | 0.914 (0.162) |
> | parp1 | 0.717 (0.167) | 0.577 (0.160) | 0.490 (0.169) | 1.000 (0.000) | 0.827 (0.160) |
> | 5ht1b | 0.690 (0.204) | 0.551 (0.200) | 0.484 (0.138) | 1.000 (0.000) | 0.801 (0.191) |
>
> From this result and the results described in Section 4.2 and Appendix A.1, we can confirm that posing an explicit constraint on pharmacochemical acceptability by removing the inappropriate fragments from the pool would be an effective strategy to generate appropriate molecules. Also, we can observe the trade-off between required constraints (realistic molecules) and uniqueness (exploration) in the result. However, our model with the large library showed moderate uniqueness, despite the strong constraint of fragment-based generation. Thus, we believe that constructing a fragment library that is large enough to guarantee high uniqueness while only including pharmacochemically acceptable fragments will be the best strategy in production.
>
> * * *
> **Q3.** On Figure 7b) (and in Figure 7a) for protein 5htb1) the results basically say that the proposed method is no more effective than attaching fragments randomly to generate higher docking molecules, which makes me wonder if using PER strategies is effective at all;
>
> **A3.** First, Figure 6a) for protein 5ht1b shows that the RL training shifted the distribution of the generated molecules towards higher docking score. As you mentioned, such an improvement is not evident in Figure 6b). However, if you observe the distribution of active molecules from Figure 6b), you can see that in 5ht1b’s case, the distribution approaches the high docking score just by adding random fragments on the scaffold. Because the influence of the scaffold on the docking score is very strong, the improvement due to the fragment attachment might not be clearly observed. On the other hand, in other targets’ cases, the effect of the RL training on the docking score distribution is evident.
> * * *
> **Q4.** How are bond types handled by the GCN? Judging from line 150, the GCN doesn't handle edge types, which is quite unlikely.
>
> **A4.** As you mentioned, our GCN doesn’t explicitly handle bond types as edge information. However, we take bond types into account by implicitly inserting the information in nodes. Bond type can be uniquely determined by the valency and number of hydrogens of the atoms that form that bond.
> * * *
>
> **Q5.** What are the novelty rates (i.e. the ratio of generated molecules that are not in the training set) of the model? Why you didn't report them?
>
> **A5.** We train our model with the RL algorithm, and our model doesn’t require pre-training. Therefore, we do not need or have any training set, and we can consider all generated molecules as novel molecules.
> * * *
>
> **Q6.** Similarly, the authors use several techniques to improve the diversity when generating molecules. So I would have expected that the chemical diversity of the 3000 sample was measured, when in fact it wasn't.
>
> **A6.**
> In theory, it is true that fragment-based molecule generation may limit the diversity of generated molecules compared to atom-based or smiles-based generation methods. However, our model grasps chemical-realisticness through fragment-based generation, while we solve the problem of diversity with explorative experience replay which we also proposed. Balancing required constraints (realistic molecules) and diversity(exploration) is the main contribution of our paper.
>
> We computed the Internal Diversity ($\text{IntDiv}_1$) of our model and the baseline models according to the widely used MOSES benchmark [1]. Observing the result below, despite the strong constraint on the molecule structure, our model shows moderate diversity. The trade-off between required constraints and diversity is also observed from comparing ours (small) and ours (large).
>
> ours (small): FREED(PE) trained with the small library used for Section 4.2.
>
> ours (large): FREED(PE) trained with the large library used for Section 4.3.
>
> **Internal diversity**
>
> |  | fa7 | parp1 | 5ht1b |
> | --- | --- | --- | --- |
> | ours (small) | 0.778 (0.021) | 0.787 (0.015) |0.782 (0.017) |
> | ours (large) | 0.837 (0.014) | 0.817 (0.018) | 0.817 (0.010) |
> | REINVENT | 0.862 (0.008) | 0.860 (0.007) | 0.859 (0.009) |
> | MORLD | 0.897 (0.009) |0.895 (0.001)  | 0.839 (0.003)|
>
> * * *
> **Q7.** When does the generative process stop? I couldn't see it explained on the paper. Perhaps I can find it in the supplementary material? It's information that I would like to read in the paper itself, though.
>
> **A7.** According to Figure 4 of (J Cheminform. 2016; 8: 14.), the most popular number of fragments in DUD-E active compounds are around 4~6. Considering that, we only considered four-action episodes in this work. We didn’t make our algorithm learn when to stop, but we leave this as our imminent future work.
> * * *
> **Q8.** One that I find critical for these types of models is computational time. It takes a long time to compute accurate docking simulations. In this paper, the authors use a docking program that is known to produce faster but less reliable docking estimations. However, no one would use this program in a practical scenario; hence, it should be properly discussed by the authors what is the computational burden to train the model for its usage in the real world, not just for experimenting.
>
> **A8.** This is a critical misunderstanding. The docking program we used in this work, QuickVina 2, is a widely used program for virtual high-throughput screening (vHTS). In big pharmaceutical companies, vHTS is often performed on a billion-molecule scale. This would cost a lot of time and computational resources if it is computed with slower but more reliable programs such as AutoDock Vina.
>
> QuickVina 2 is known to provide 20.39-fold acceleration compared to AutoDock Vina according to the paper [2]. Also, according to the paper, the binding affinity predictions from QuickVina 2 and AutoDock Vina show Pearson's correlation coefficient of 0.911. Since 0.911 indicates a high correlation, we believe that using QuickVina 2 instead of AutoDock Vina would not seriously harm the fidelity.
>
> * * *
> [1] Polykovskiy, Daniil, et al., "Molecular Sets (MOSES): A Benchmarking Platform for Molecular Generation Models", Frontiers in Pharmacology, 2020
>
> [2] Alhossary, Amr, et al., "Fast, accurate, and reliable molecular docking with QuickVina 2", Bioinformatics, Volume 31, Issue 13, 1 July 2015, Pages 2214–2216

---

> > ### Comment · Reviewer_TmJ6 · 2021-08-28
> > **Response to rebuttal**
> >
> > I thank the authors for clarifying many things (such as the choice of using QuickVina instead of AutoDock Vina in A8, which I clearly misinterpreted, or the response about the novelty rates in A5), and for having added some results regarding the diversity of the generated molecules (A6). As regards A3, I believe this explanation is crucial and should be added to the text, since it's hard to grasp by oneself.
> >
> > Taking for granted that the authors will make serious proofreading to fix the language, I still have doubts that won't make me tend towards acceptance. First, as reviewer bYsg points out, there are still some claims that are not supported by empirical evidence. Second, I had the same impression as reviewer USDx about the rushiness and sort of incompleteness of this work (see A7), which doesn't make me appreciate the technical contribution to the fullest.
> >
> > Overall, I'm raising my score to 5, but I'm still leaning towards rejection. Nonetheless, I believe this paper has value, and I encourage the authors to resubmit to other conferences once the proper adjustments are made.

---

> > > ### Author Response · Authors · 2021-09-01
> > > **Response**
> > >
> > > Thank you very much again for your time and efforts in reviewing our paper. We will do our best to improve the quality of our writing, clarify the incomplete descriptions, and supplement the experimental evidence in the revision.

---

### Official Review · Reviewer_5BYY · 2021-07-17

**Rating:** 4
**Confidence:** 4

**Summary:**

The authors propose a novel approach based on reinforcement learning (RL) to accelerate the discovery of pharmacologically relevant inhibitors. One of the key limitations of existing methods is that many molecules produced by generative models based are chemical invalid. In this work, the authors circumvent this problem by allowing the agent to modify an existing ligand (state) with a set of actions that result in a new chemically valid molecule (next state). Through this introduction of domain knowledge, the authors achieve higher sample efficiency. However, the experiments show that this inadvertently reduces the diversity of generated molecules.

Overall, the approach seems reasonable, and experimental results impressive. However, the manuscript lacks methodological novelty, and thus, might not be suited for this venue. Further, the presentation (language, precision, and figures) requires work. For this reason, I vote for rejection.

**Limitations And Societal Impact:**

1. The authors need to revise the manuscript to improve readability and precision. Currently, the manuscript is not ready for publication.
1. The proposed method seems reasonable, but the novelty on the machine learning side is very limited. The contributions are worthwhile but most probably not relevant to the target audience of this venue.

**Main Review:**

# Abstract
- L7: define pharmacochemically acceptable more precisely
- L8: "the docking score optimization is a difficult exploration problem that involves many local optima and less smooth surface". This statement is not clear
- L13: What is "qualified chemical space"?

# Introduction
- L30: is there is a citation missing?
- Similar to the abstract, the introduction requires more precision.

# Related Work
- Good overview of existing methods but no mention of RL approaches that generate 3D structures, such as:
    - Meldgaard, S. A.; Mortensen, H. L.; Jørgensen, M. S.; Hammer, B. Structure Prediction of Surface Reconstructions by Deep Reinforcement Learning. J. Phys.: Condens. Matter 2020, 32 (40), 404005. https://doi.org/10.1088/1361-648X/ab94f2.
    - Simm, G. N. C.; Pinsler, R.; Hernández-Lobato, J. M. Reinforcement Learning for Molecular Design Guided by Quantum Mechanics. In Proceedings of the 37th international conference on machine learning; III, H. D., Singh, A., Eds.; Proceedings of machine learning research; PMLR, 2020; Vol. 119, pp 8959–8969.
    - Christiansen, M.-P. V.; Mortensen, H. L.; Meldgaard, S. A.; Hammer, B. Gaussian Representation for Image Recognition and Reinforcement Learning of Atomistic Structure. J. Chem. Phys. 2020, 153 (4), 044107. https://doi.org/10.1063/5.0015571.
    - Simm, G. N. C.; Pinsler, R.; Csányi, G.; Hernández-Lobato, J. M. Symmetry-Aware Actor-Critic for 3D Molecular Design; ICLR 2020.

**Time Spent Reviewing:**

3

---

> ### Author Response · Authors · 2021-08-10
> **Response**
>
> We sincerely appreciate your constructive and helpful comments. We apologize for the writing quality of our initial manuscript. We have proofread and revised our manuscript to improve the readability of our manuscript.
>
> **Q1.** The manuscript lacks methodological novelty, and thus, might not be suited for this venue.
>
> **A1.** We strongly disagree with the claim that our work lacks methodological novelty. Our design of the generative procedure, the model architecture, and the PER + SAC algorithm to support the explorative search of the potential drugs are unique, and our design is shown to be highly effective. We list the main contributions in methodology which the readers of this venue will be interested in as follows:
>
> - Our proposed method aims to solve practical yet rarely noticed problems in data-driven drug discovery. Various real-world AI application problems face critical practical challenges that do not exist in synthetic problems, and our work’s contribution is in the creative combination of the new generative algorithm design and the machine-learning techniques.
> - First, we devised methods that can integrate expert knowledge into data-driven drug discovery. Our design of fragment-based molecule generation enables the practitioner to constrain the generated molecules to be objective-specific by controlling the fragment library. Also, taking the common attachment sites in fragments as potential connection points are very relevant in drug design. Such modifications are closely related to the design of the actions and the states of RL, which will attract the attention of the readers of this venue.
> - Second, we devised a new Prioritized Experience Replay (PER) algorithm called PER(PE), where PE stands for predictive error. In previous works that used the PER algorithm, priority was typically obtained with TD error. In our work, we introduce the usage of error of the reward predictor as a sample priority and confirm the effectiveness with ablation studies. This modification in the algorithm was essential in solving the main problem -- finding as many hit molecules by enhancing RL exploration.
>
> * * *
> **Q2.** Readability concerns.
>
> **A2.** Following your suggestion, we have corrected the grammatical errors and notations. We will further work on improving the quality of the writing for the revision, by spending significant time and effort on revising the paper. We sincerely thank you for pointing this out.
> * * *
> **Q3.** Related works.
>
> **A3.** Thank you for suggesting highly relevant works. We will discuss these works in the revision.

---

### Official Review · Reviewer_USDx · 2021-07-19

**Rating:** 5
**Confidence:** 2

**Summary:**

The authors present a reinforcement learning framework for the design of molecules active against specific proteins. The novelty in the authors methodology is to attach molecular fragments (as opposed to single atoms) at each step and optimise the algorithm with respect to the molecular docking score. Attaching fragments allows for the generation of pharmacochemically-valid molecules (those which are suitable drug-candidates) by only attaching pharmacochemically-valid attachments. Optimising with respect to the docking score allows the algorithm to attain a good proxy of the binding affinity of a given molecule to a given protein.

Given a current state (the current molecule), the framework designs molecules in three steps:

1) The optimal attachment site is identified on the current molecule.
2) Given the choice of site, the optimal molecular fragment is chosen from a set of predefined fragments.
3) Given the choice of site and fragment, the optimal attachment site on the fragment is identified.

The experimental results indicate that the proposed approach generates many valid molecules in comparison to baseline approaches.


**Ethical Concerns:**

Section 6 covers this sufficiently.

**Limitations And Societal Impact:**

Sections 5 & 6 cover this sufficiently.

**Main Review:**

Originality:
I believe the work possesses originality in the way the authors have designed their framework. The novelty in the authors methodology is to attach molecular fragments (as opposed to single atoms) at each step and optimise the algorithm with respect to the molecular docking score. Attaching fragments allows for the generation of pharmacochemically-valid molecules (those which are suitable drug-candidates) by only attaching pharmacochemically-valid attachments. Optimising with respect to the docking score allows the algorithm to attain a good proxy of the binding affinity of a given molecule to a given protein.

Quality:
Whilst the results look promising, I think there are areas for improvement. The standard of writing could be improved as it can be confusing at times. The interpretation of the results seems shallow.

For instance, why ignore the very large standard deviations in figure 4? The authors state (lines 250-256) "Our model outperforms the other generative models MORLD and REINVENT in terms of both hit ratio and top 5% score, except for the hit ratio of the 5ht1b case. Such results show that our model’s performance is superior or at least competitive to existing base lines...". On average, this does happen. Yet, the large variability indicates that it is not always the case. This deserves further investigation, surely? Either through more random restarts or at least a plausible explanation of why this variability occurs. On the topic of figure 4, it does not indicate which plots correspond to which proteins. Also, I think a description of how to read the top 5% plots would be helpful (I can only assume the greater negative scores indicate better performance?).

The interpretation of Figure 7 is that the molecules look "chemically realistic". I find this phrase somewhat misleading since, to me (I'm not a chemist), any two-dimensional molecular graph looks chemically realistic. What makes these specific structures look realistic? I understand the authors want to show off the molecules they have generated (rightly so!) but they must be careful with their interpretations.

Clarity:
This is mentioned in the previous section but the standard of writing could be improved. The work seems a little rushed, which is a shame since it diverts the attention from the nice results.

Significance:
Machine learning applied to drug-design is currently a fairly hot topic, so this paper is of obvious significance.

General remarks:
The paper makes a nice contribution through the use of fragment-based drug design and the results mainly back this up (see my comment regarding  Figure 4). However, the work seems more like a first draft than a submission-ready paper. For this reason, I would have to reject.


**Time Spent Reviewing:**

4

---

> ### Author Response · Authors · 2021-08-09
> **Response**
>
> We sincerely appreciate your constructive and helpful comments. We apologize for the writing quality of our initial manuscript. Following your suggestion, we have corrected the grammatical errors and notations. We will further work on improving the quality of the writing for the revision, by spending significant time and effort on revising the paper. We sincerely thank you for pointing this out.
>
> We initially address all your comments below:
>
> **Q1.** Why ignore the very large standard deviations in figure 4?
>
> **A1.**
> We performed a two-sided Welch’s t-test on hit ratio and the top 5% score of our model and baseline models. The null hypothesis is that 2 independent samples have identical average (expected) values. (Hypothesis: $\mu_1 -  \mu_2 = 0 $)
>
> The result (p-value) is shown below. From the p < 0.05 standard, we can see that FREED(PE) shows significantly better results compared to MORLD in terms of hit ratio and top 5% score. Also, FREED(PE) shows significantly better results compared to REINVENT in terms of the top 5% score. Although our model is not significantly superior to REINVENT, we acknowledged the fact ("superior or at least competitive") and claimed our model's unique practical advantages in lines 250-256.
>
> **hit ratio**
>
> | t-statistic(p-value) | fa7 | 5ht1b | parp1 |
> | --- | --- | --- | --- |
> | FREED(PE) - REINVENT | 1.8193(0.1401) | -1.5312(0.1863) | 0.6957(0.5217) |
> | FREED(PE) - MORLD |3.0177(0.0392) | 3.4127(0.0264) | 4.72(0.0088) |
>
> **top 5% score**
>
> | t-statistic(p-value) | fa7 | 5ht1b | parp1 |
> | --- | --- | --- | --- |
> | FREED(PE) - REINVENT | 2.68(0.0482) | 3.4095(0.0157) | 2.7566(0.0371) |
> | FREED(PE) - MORLD | 6.4538(0.0025) | 10.601(0.0004) | 8.2617(0.0008) |
>
> * * *
> **Q2.** On the topic of figure 4, it does not indicate which plots correspond to which proteins. Also, I think a description of how to read the top 5% plots would be helpful (I can only assume the greater negative scores indicate better performance?).
>
> **A2.** We apologize for the confusion, and we thank the reviewers for pointing this out. We have revised the manuscript for clarity.
>
> * * *
> **Q3.** The interpretation of Figure 7 is that the molecules look "chemically realistic". I find this phrase somewhat misleading since, to me (I'm not a chemist), any two-dimensional molecular graph looks chemically realistic. What makes these specific structures look realistic? I understand the authors want to show off the molecules they have generated (rightly so!) but they must be careful with their interpretations.
>
> **A3.** Thank you for pointing this out, and we will strengthen our descriptions in the revision. We widely define the term “chemically realistic molecule” as a stable molecule and narrowly define it as a molecule that is made up of fragments that appear in the ZINC data. Also, we widely define the term “inappropriate molecule/fragment” or “pharmacochemically inacceptable molecule/fragment” as a molecule that has nonspecific toxicity or reactivity, and narrowly define it as a molecule that cannot pass through the three medicinal chemistry filters, which are Glaxo, PAINS, SureChEMBL filters.
>
> We also report the SA (synthetic accessibility) scores [1] and steric strain filter [2] results here, and we will also add this information in the revision. SA score is a widely used metric that estimates ease of synthesis by penalizing the presence of non-standard structural features. We applied the same steric strain filter that was introduced in [2], which returns False if a molecule is too sterically strained so that the average angle bend energy computed with MMFF94 forcefield minimization method exceeds a cutoff of 0.82 kcal/mol.
>
> According to Figure 4 of [1], the SA score distribution of the catalogue molecules of commercial compound providers has its mode around 3. Accordingly, we can think of our generated molecules as fairly synthesizable and thus chemically realistic. Also, all molecules passed the steric strain filter, meaning that the steric strains are not strong in our molecules, thus being stable.
>
> The label (e.g. (i) (a)) follows the way we labeled the molecules for Figure 7.
>
> |  | SA score | Steric strain filter |
> | --- | --- | --- |
> |(i) (a) | 3.50 | True |
> |(ii) (a) | 3.78 | True |
> |(i) (b) | 3.89 | True |
> |(ii) (b) | 3.75 | True |
> |(i) (c) | 2.54 | True |
> |(ii) (c) | 5.42 | True |
>
> * * *
> [1] Ertl, Peter, et al., "Estimation of synthetic accessibility score of drug-like molecules based on molecular complexity and fragment contributions",  Journal of Cheminformatics volume 1, Article number: 8 (2009)
>
> [2] You, Jiaxuan, et al., "Graph Convolutional Policy Network for Goal-Directed Molecular Graph Generation" arXiv preprint arXiv:1806.02473 (2018)

---

> > ### Comment · Reviewer_USDx · 2021-08-31
> > **Reply**
> >
> > I think the information provided in A1 & A3 should certainly be added to the work as they help clarify my initial confusion. I still have concerns regarding the writing style, but if the authors were to fix everything and improve the general readability then I would be happy to raise my score to a 6.

---

> > > ### Author Response · Authors · 2021-09-01
> > > **Response**
> > >
> > > We will do our best to improve the readability and clarity of our manuscript in the revision. We will also add suggested additional experimental results to our manuscript. Thank you very much again for your time and efforts in reviewing our paper.

---

### Decision · Program_Chairs · 2021-09-28

**Decision:**

Accept (Poster)

**Comment:**

The submission has its merits but, even after carefully checking the author rebuttal, the reviewers unanimously agree that the submission is not strong enough for publication at NeurIPS. The main reasons for that are the lack of novelty, clarity, and supporting evidence.


**Consistency Experiment:**

NeurIPS has a long history of experimentation. In 2014, NeurIPS ran an experiment in which 10% of submissions were reviewed by two independent committees to quantify the randomness in the review process. This year, we repeated a variant of this experiment to see how the quality of the review process has changed over time.  This paper was part of the experiment and was therefore assigned to two committees (consisting of reviewers, an Area Chair, and a Senior Area Chair) that reached independent decisions.  If both committees made the same recommendation, this recommendation was followed. If a single committee recommended acceptance, the paper was accepted (with the exception of a few cases in which the other committee identified what we considered a fatal flaw, e.g., an error in a key result).

This copy’s committee reached the following decision: **Reject**

The other committee assigned to the paper recommended **Accept (Poster)**.  You can find the other set of reviews, along with any follow up discussion with the authors here:
https://openreview.net/forum?id=Msc9XKd-3bA